# ENHANCING INSTRUCTION FOLLOWING OF LLMS VIA ACTIVATION STEERING WITH DYNAMIC REJECTION

**Minjae Kang**[1]    **Jaehyung Kim**[1]
[1]Yonsei University
{mjkang618,jaehyungk}@yonsei.ac.kr

## ABSTRACT

Large Language Models, despite advances in instruction tuning, often fail to follow complex user instructions. Activation steering techniques aim to mitigate this by manipulating model internals, but have a potential risk of *oversteering*, where excessive emphasis on the instruction degrades task accuracy and overall text quality. To address this, we introduce **DIRECTER** (**D**ynamic **re**jection s**teer**ing), a novel steering method that dynamically modulates steering strength by scaling the KV cache without extra dataset. DIRECTER couples steering with a *plausibility-guided* decoding loop, which adaptively adjusts steering strength at each step by comparing the steered output distribution to the original. If the steered output is deemed implausible, steering strength is progressively weakened. This strength modulation is guided by a lightweight, one-time *attention sensitivity analysis* that ranks layers by their influence on model representations. Extensive evaluations show that DIRECTER significantly enhances instruction-following capabilities across diverse benchmarks, improving accuracy by up to 6.5% over baselines without the common trade-offs in generation quality or task fidelity. The proposed dynamic, plausibility-guided control during activation steering further demonstrates its potential as a general mechanism for mitigating oversteering that is compatible with existing baselines.

## 1 INTRODUCTION

Large Language Models (LLMs) (Brown et al., 2020; Dubey et al., 2024) have achieved unprecedented advancements in recent years, demonstrating expert-level performance across diverse domains from code generation (Chen et al., 2021) to research assistance (Lu et al., 2024). This remarkable progress has been largely enabled by *instruction tuning* (Ouyang et al., 2022; Rafailov et al., 2023; Bai et al., 2022a), a post-training paradigm designed to bridge the gap between a generic next token prediction objective at pre-training and end-user requirements by aligning model behavior with human preferences and task-specific instructions. While instruction tuning has proven effective (Qin et al., 2024; Asai et al., 2024), solely relying on this training-based approach has a certain limitation to fully cover the vast diversity of real-world user instructions (Koh et al., 2021).

One strategy to address this limitation is *activation steering* (Li et al., 2023b; Panickssery et al., 2023; Zou et al., 2023; Turner et al., 2023), which aims to improve instruction-following capabilities at inference time by steering the internal activations of LLMs. For example, PASTA (Zhang et al., 2023) first profiles attention heads to identify those improving task performance, then suppresses attention on non-instruction tokens within these selected heads during inference. In contrast, SpotLight (Venkateswaran & Contractor, 2025) amplifies the attention scores corresponding to the instruction tokens, ensuring consistent attention mass allocation to the instruction. However, such approaches have a common potential risk of *oversteering*, where an excessive emphasis on the instruction often comes at the cost of task accuracy and can also degrade the overall quality of the generated text (Bi et al., 2024; Hedström et al., 2025; Belitsky et al., 2025). Addressing this risk is challenging as these methods often rely on manually-tuned hyperparameters; this static approach incurs search costs and fundamentally fails to adapt to the optimal degree of steering that dynamically changes at each decoding step (Lee et al., 2024; Wang et al., 2025; Postmus & Abreu, 2024).

---

Implementation code is available at https://github.com/mjk0618/directer.

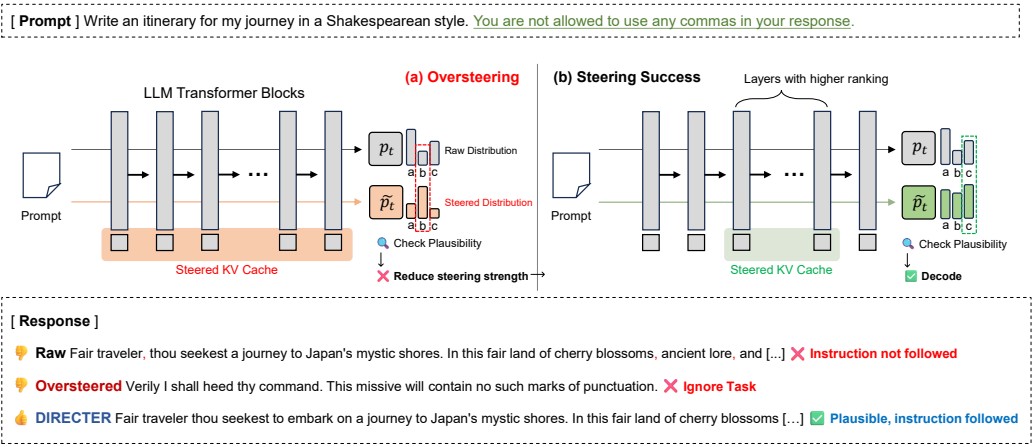

**Figure 1: An overview of DIRECTER's plausibility-guided decoding loop.** At each step, a steered output distribution ($\tilde{p}_t$) from KV cache scaling is compared against the raw output distribution ($p_t$). **(a) Steering Failure:** If the steered candidate is deemed implausible, it is rejected, triggering a progressive reduction of steered layers to weaken the intervention. **(b) Steering Success:** If the candidate is plausible, it is accepted for decoding.

**Contribution.** In this work, we propose a new steering method that mitigates oversteering risk via a novel **D**ynamic **re**jection s**teer**ing mechanism (DIRECTER). Our key idea is to prevent oversteering by automatically modulating the steering strength at each decoding step. Specifically, DIRECTER integrates steering with a *plausibility-guided* decoding loop (Li et al., 2022; Chuang et al., 2023). At each step, DIRECTER tentatively applies KV cache scaling to all layers and compares the output distributions from the steered and raw forward passes. The steering is accepted only if the steered output distribution does not deviate excessively from the raw distribution. If it does, DIRECTER progressively reduces the number of steered layers to weaken the steering strength until the deviation is acceptable. For fine-grained control of steering strength, we further introduce a layer ranking mechanism based on attention sensitivity, which measures the distributional shift propagated through attention layers from a single-layer steering, the effectiveness of which we validate empirically.

Our extensive empirical evaluations on diverse benchmarks, including IFEval (Zhou et al., 2023), show that DIRECTER improves average accuracy by 6.5% over the baseline and surpasses prior steering methods by approximately 4%. Crucially, unlike prior steering methods, this gain does not sacrifice task correctness or text quality. DIRECTER achieves the highest task fidelity ($\approx$92%) in LLM-judged evaluations while matching the text quality of non-intervention baselines. This gain is also achieved with practical efficiency, as our gating mechanism limits the throughput reduction to a modest $\approx$16% with negligible memory overhead. Furthermore, our plausibility-guided decoding serves as a general mechanism to mitigate oversteering, boosting the performance of other methods when applied as a safety gate. These results establish DIRECTER as a practical and mechanistic method for achieving more reliable and controllable LLM generation.

## 2 RELATED WORK

**Instruction following of LLMs.** LLMs are pre-trained for simple next-token prediction, which is misaligned with the end-user goal of producing responses that adhere to nuanced instructions. Instruction tuning helps bridge this gap by aligning model behavior with human preferences and task-specific instructions (Ouyang et al., 2022; Rafailov et al., 2023; Bai et al., 2022a), yet the vast diversity of real-world instructions makes it inherently challenging to cover all scenarios through training alone (Koh et al., 2021; Bukharin & Zhao, 2024). Prompting-based alternatives at inference time, such as augmenting prompts with few-shot task demonstrations or using structured prompts that explicitly emphasize the rules to be followed, can improve performance (Chang et al., 2024; Liu et al., 2021). However, these approaches often remain brittle for fine-grained constraints and may degrade as prompt length increases (Holtzman et al., 2019; Liu et al., 2023a; Li et al., 2023a).

These limitations motivate *activation steering* methods that act on the model's inference process itself, steering how the prompt is understood and processed.

**Activation steering.** Activation steering approaches manipulate the internal model states at inference time, aiming to make models more controllable and reliable (Li et al., 2023b; Panickssery et al., 2023; Zou et al., 2023; Turner et al., 2023). Early approaches injected steering vectors into residual streams using contrastive prompt pairs, but required many carefully designed pairs and extensive sweeps over layers/heads (Turner et al., 2023; Chen et al., 2025). Recent attention-level steering methods address this by operating directly on attention distributions; PASTA profiles attention heads that empirically benefit task performance and suppresses attention to non-instruction tokens, but demands hundreds to thousands of validation examples and exhaustive grid search across layer–head combinations, costs comparable to training-level pre-computation (Zhang et al., 2023; 2024). In contrast, SpotLight (Venkateswaran & Contractor, 2025) adjusts attention to maintain a target proportion on instruction tokens via post-softmax logit biasing at each decoding step, improving adherence but effectively doubling softmax operations and increasing latency. More fundamentally, many activation steering methods rely on fixed configurations throughout generation, failing to capture the dynamics of text generation; they are hyperparameter-sensitive and prone to *oversteering*, where emphasizing the instruction degrades overall text quality (Bi et al., 2024; Hedström et al., 2025; Belitsky et al., 2025; Lee et al., 2024; Wang et al., 2025; Postmus & Abreu, 2024). In contrast to prior methods, our work, DIRECTER, introduces a dynamic control mechanism that couples KV cache steering with a plausibility-guided decoding, allowing it to adaptively modulate steering strength at each step to enhance instruction following of LLMs without sacrificing generation quality.

## 3 DIRECTER: DYNAMIC REJECTION STEERING TO FOLLOW INSTRUCTIONS

### 3.1 PRELIMINARIES

**Attention mechanism.** We consider a standard $L$-layer decoder-only Transformer (Vaswani et al., 2017) with hidden states of dimension $d$. At each decoding step $t$, the model processes an input sequence of tokens $x_{1:t-1}$ to sequentially produce a hidden state $\mathbf{H}^{(l)}$ at each layer $l$. Within $l$-th layer, the input hidden state $\mathbf{H}^{(l-1)} \in \mathbb{R}^{(t-1) \times d}$ is projected into queries ($\mathbf{Q}^{(l)}$), keys ($\mathbf{K}^{(l)}$), and values ($\mathbf{V}^{(l)}$) using corresponding weight matrices $\mathbf{W}_Q^{(l)}, \mathbf{W}_K^{(l)}, \mathbf{W}_V^{(l)} \in \mathbb{R}^{d \times d}$. The self-attention output is then computed as:

$$\text{Attn}(\mathbf{Q}^{(l)}, \mathbf{K}^{(l)}, \mathbf{V}^{(l)}) = \text{softmax}(\mathbf{S}^{(l)})\mathbf{V}^{(l)}, \quad \text{where} \quad \mathbf{S}^{(l)} = \mathbf{Q}^{(l)}(\mathbf{K}^{(l)})^\top/\sqrt{d},$$

with the softmax function applied in a row-wise sense. We denote the hidden states immediately before and after this attention sub-block as $\mathbf{H}_{\text{pre}}^{(l)}$ and $\mathbf{H}_{\text{post}}^{(l)}$, respectively (*i.e.*, $\mathbf{H}_{\text{pre}}^{(l)} := \mathbf{H}^{(l-1)}$ and $\mathbf{H}_{\text{post}}^{(l)} := \text{Attn}(\mathbf{Q}^{(l)}, \mathbf{K}^{(l)}, \mathbf{V}^{(l)})$). The final layer's output, $\mathbf{H}^{(L)}$, is projected to a logit vector $\boldsymbol{\ell}_t$ to produce the next-token probability distribution $p_t = \text{softmax}(\boldsymbol{\ell}_t)$. Henceforth, unless otherwise specified, we use the term *layer* to refer specifically to the self-attention sub-block, excluding the feed-forward network and residual connections.

**KV cache.** For efficient autoregressive generation, past key and value vectors are cached (Pope et al., 2023). It begins with a *prefill* phase, where the KV cache is populated for the input prompt $x_{1:T}$. Then, at each subsequent decoding step $t > T$, the model computes a new key vector $\mathbf{k}_t^{(l)}$ and value vector $\mathbf{v}_t^{(l)}$ for the current token. These are appended to the existing cache for each layer $l$:

$$\mathbf{K}_{1:t}^{(l)} = [\mathbf{K}_{1:t-1}^{(l)}; \mathbf{k}_t^{(l)}]; \quad \mathbf{V}_{1:t}^{(l)} = [\mathbf{V}_{1:t-1}^{(l)}; \mathbf{v}_t^{(l)}].$$

The attention output is then computed using the current token's query vector, $\mathbf{q}_t^{(l)}$, and the full updated KV cache, $\{\mathbf{K}_{1:t}^{(l)}, \mathbf{V}_{1:t}^{(l)}\}$.

**Activation steering.** Activation steering modifies the model's internal representations during inference. We define the user-specified *instruction span* by the token indices $\mathcal{I} \subseteq \{1, \ldots, T\}$ and the selected steering layers as $\mathcal{L} \subseteq \{1, \ldots, L\}$. This can be done at the attention level by adding a bias

$b_{t,i}$ to $\mathbf{S}$, for instance: $\mathbf{S}'_{t,i} = \mathbf{S}_{t,i} + b_{t,i} \cdot \mathbb{1}[i \in \mathcal{I}]$. In contrast, KV cache steering directly modifies the key or value vectors. Given scaling factors $\alpha_K$ and $\alpha_V$, the general form of this intervention is:

$$\mathbf{k}'^{(l)}_i = \begin{cases} \alpha_K \cdot \mathbf{k}^{(l)}_i, & \text{if } i \in \mathcal{I} \text{ and } l \in \mathcal{L} \\ \mathbf{k}^{(l)}_i, & \text{otherwise} \end{cases} ; \quad \mathbf{v}'^{(l)}_i = \begin{cases} \alpha_V \cdot \mathbf{v}^{(l)}_i, & \text{if } i \in \mathcal{I} \text{ and } l \in \mathcal{L} \\ \mathbf{v}^{(l)}_i, & \text{otherwise} \end{cases}$$

In this work, we focus on key scaling.[1] Namely, we fix the value scaling factor $\alpha_V = 1$, and unless otherwise stated, use $\alpha$ to denote the key scaling factor $\alpha_K \geq 1$. While steering strength can be controlled by either the scaling factor $\alpha$ or the set of steered layers $\mathcal{L}$, we primarily focus on modulating $\mathcal{L}$, the reason for which is detailed in Appendix C.4.

## 3.2 KV Cache Steering with Plausibility Constraint and Layer Ranking

To overcome the limitations of static activation steering methods, which can often lead to oversteering, we propose **DIRECTER** which adaptively adjusts steering strength at each generation step to dynamically balance instruction following and task performance. For a high-level overview and the overall procedure, see Figure 1 and Algorithm 1, respectively. DIRECTER mitigates oversteering through a **plausibility-guided decoding** loop. For effective control of steering strength, it further employs a **layer ranking with attention sensitivity** to determine the number of layers to be steered.

**Plausibility-guided decoding.** We first perform a standard forward pass at each decoding step to obtain the raw output probability distribution $p_t$. We then initiate a progressive steering loop with a candidate set of steered layers, $\mathcal{L}_{\text{cand},t}$, initialized from our ranked list $\mathcal{L}_{\text{ranked}}$ (detailed in Eq. 3). This process yields a steered output distribution $\widetilde{p}_t$. Instead of naively using $\widetilde{p}_t$, we check its plausibility. Let $i_t^*$ and $\widetilde{i}_t^*$ be the top-1 token indices of the distributions $p_t$ and $\widetilde{p}_t$, respectively. The steered distribution is accepted only if its new top token, $\widetilde{i}_t^*$, was considered sufficiently plausible by the original distribution $p_t$. Formally, the intervention is accepted if:

$$p_{t,\widetilde{i}_t^*} \geq \beta \cdot p_{t,i_t^*} \tag{1}$$

where $\beta \in [0,1]$ is a plausibility threshold. If this condition is not met, we progressively halve the candidate set (*i.e.*, $|\mathcal{L}_{\text{cand},t}| \leftarrow \lfloor |\mathcal{L}_{\text{cand},t}|/2 \rfloor$), removing those with the lowest sensitivity.[2] This cycle continues until a steered prediction is accepted or the set $\mathcal{L}_{\text{cand},t}$ becomes empty. If no steered distribution passes this filter, we use the raw distribution $p_t$ to generate the next token.

While effective, the recursive decoding loop with plausibility guidance introduces computational overhead. To mitigate this, we introduce an efficient gating mechanism that pre-determines when a steered forward pass can be safely skipped. The decision is based on the probabilities of the top-2 tokens from the original distribution, denoted as $p_{t,i_t^*}$ and $p_{t,i_t^{**}}$, respectively. If $p_{t,i_t^{**}} < \beta \cdot p_{t,i_t^*}$, we can guarantee that no steered distribution $\tilde{p}_t$ could satisfy the plausibility constraint in Eq. 1 except when $\widetilde{i}_t^* = i_t^*$ (*i.e.*, same top-1 predictions). In such cases, we skip the steering attempt and use the original prediction $p_t$, significantly reducing computational cost without performance degradation.

**Layer ranking with attention sensitivity.** To enable a principled reduction of steering strength, we perform a one-time layerwise sensitivity analysis before starting the decoding. Our approach inverts the logic of KV cache quantization (Ge et al., 2023; Wang et al., 2024b); whereas quantization identifies low-impact layers for compression (*i.e.*, high input-output similarity), we find the most influential layers by measuring the deviation after the steering.

To be specific, the analysis steers one layer $\ell$ at a time by scaling its key vectors of instruction tokens. The resulting impact on each layer $j$ is quantified by a **disturbance score**, $D_j(\ell)$. This score measures the deviation relative to the original representational shift by attention layer from a

---

[1] We focus on key scaling for efficiency. Its effect is naturally renormalized by the subsequent softmax function, whereas value scaling requires extra computation for an explicit renormalization step. Our empirical results also confirm that key scaling is more effective (see Appendix C.5).

[2] We found it is more effective to reduce steering strength by gradually removing the least sensitive layers first. Conversely, when the ranking is reversed, steering is applied mainly to the least sensitive layers, which is often insufficient to produce a token-level effect and causes the output to default to the raw distribution.

---

**Algorithm 1:** DIRECTER Algorithm Overview

---

**Input:** Prompt $\mathbf{x}$, plausibility threshold $\beta$
**Output:** Generated sequence $y$

1 Prefill $\mathbf{x}$ to obtain the original KV cache $\mathcal{C}$.
2 Run a one-time attention sensitivity analysis to get ranked layers $\mathcal{L}_{\text{ranked}}$. (Eq. 2, 3)
3 Initialize $y \leftarrow [\,]$.
4 **while** *not end-of-sequence* **do**
5      Run a raw forward pass to obtain $p_t = \text{softmax}(\ell_t)$ and the top-2 probability tokens $(i_t^*, i_t^{**})$.
6      **if** $p_{t,i_t^{**}} < \beta \cdot p_{t,i_t^*}$ **then**
7          $p_{\text{final}} \leftarrow p_t$ // `skip steering`
8      **else**
9          Initialize $\mathcal{L}_{\text{cand}} \leftarrow \mathcal{L}_{\text{ranked}}$ and accepted $\leftarrow$ False.
10          **while** $|\mathcal{L}_{cand}| > 0$ **and** $\neg$ accepted **do**
11              Apply steering on $\mathcal{L}_{\text{cand}}$ to obtain $\tilde{p}_t$, and set $\tilde{i}_t^* = \arg\max_i \tilde{p}_{t,i}$.
12              **if** $p_{t,\tilde{i}_t^*} \geq \beta \cdot p_{t,i_t^*}$ **then**
13                  accepted $\leftarrow$ True
14              **else**
15                  $k \leftarrow \lfloor |\mathcal{L}_{\text{cand}}|/2 \rfloor$.
16                  $\mathcal{L}_{\text{cand}} \leftarrow \mathcal{L}_{\text{cand}}[1{:}k]$ // `remove the lowest-sensitivity half`
17          $p_{\text{final}} \leftarrow \tilde{p}_t$ if accepted, otherwise $p_{\text{final}} \leftarrow p_t$.
18      Decode next token, update $\mathcal{C}$, and append to $y$.
19 **return** $y$

---

standard forward pass without steering, and isolates two effects: (1) the direct impact on layer $j$'s attention output, and (2) the propagated impact on layer $j$'s attention input. The formulation is:

$$D_j(\ell) = \underbrace{\left(\text{dist}(\mathbf{H}_{\text{pre}}^{(j)}, \mathbf{H}_{\text{post}}^{(j,\ell)}) - \text{dist}(\mathbf{H}_{\text{pre}}^{(j)}, \mathbf{H}_{\text{post}}^{(j)})\right)}_{\text{Direct effect on layer } j} + \underbrace{\left(\text{dist}(\mathbf{H}_{\text{pre}}^{(j,\ell)}, \mathbf{H}_{\text{post}}^{(j)}) - \text{dist}(\mathbf{H}_{\text{pre}}^{(j)}, \mathbf{H}_{\text{post}}^{(j)})\right)}_{\text{Propagated effect from layer } \ell} \quad (2)$$

where $\text{dist}(\cdot, \cdot)$ is the cosine distance, and $\mathbf{H}^{(j)}$ and $\mathbf{H}^{(j,\ell)}$ are the hidden states at layer $j$ obtained from the raw and steered forward passes, respectively. Then, the final **attention sensitivity** for layer $\ell$ is the average disturbance across all layers:

$$\text{Sensitivity}(\ell) = \frac{1}{L} \sum_{j=1}^{L} D_j(\ell) \quad (3)$$

Using this sensitivity score, we measure the rankings $\mathcal{L}_{\text{ranked}}$, which is generated once after the prompt prefill with minimal overhead (Section 4.3) and guides the adaptive steering process. A detailed justification for this metric's formulation and its simplification for efficient computation is provided in Appendix C. The full procedure is detailed in Algorithm 2.

## 4 EXPERIMENTS

In this section, we conduct a comprehensive set of experiments to validate our proposed method, DIRECTER. Our evaluation is designed to answer the following key research questions:

- **RQ1:** Does DIRECTER improve instruction-following across diverse benchmarks? (Table 1)
- **RQ2:** Does DIRECTER generalize to different architectures and scales? (Table 2)
- **RQ3:** Does plausibility guidance mitigate oversteering in other steering methods? (Figure 2)
- **RQ4:** Is our attention-sensitivity ranking an effective layer selection strategy? (Table 3)
- **RQ5:** Is DIRECTER efficient in terms of latency and memory overhead? (Figure 4)

### 4.1 EXPERIMENTAL SETUPS

**Evaluation datasets and metrics.** We evaluate DIRECTER on a diverse set of instruction-following benchmarks. (1) First, for strict instruction following, we use *IFEval* (Zhou et al., 2023),

Table 1: **Main results.** Accuracy (%) for different methods on IFEval (Prompt/Instruction), LIF-Bench (List/OneDoc/MultiDoc), and GSM8K-Format (Format/Task). The best and second best scores are highlighted in **bold** and underline, respectively.

| | Method | IFEval P. Acc / I. Acc | | LIFBench List / OD / MD | | | GSM8K-Format F. Acc / T. Acc | | All (Avg.) |
|---|---|---|---|---|---|---|---|---|---|
| **Baseline** | Zero-shot | 73.5 / 81.5 | | 63.4 / 68.6 / 40.9 | | | 79.2 / 82.7 | | 70.0 |
| **Prompting** | *-marked | 75.3 / 82.1 | | 64.3 / 66.9 / 44.9 | | | 83.1 / 82.9 | | 71.4 |
| | "-marked | 72.7 / 80.8 | | 63.4 / 69.9 / 41.0 | | | 77.5 / 84.0 | | 69.9 |
| | Few-shot | 74.8 / 82.2 | | 55.5 / 57.7 / 42.2 | | | 98.9 / **87.1** | | 71.2 |
| **Steering** | PASTA | 66.7 / 75.5 | | 61.1 / 62.8 / 22.5 | | | **99.2** / 48.1 | | 62.3 |
| | PASTA* | 76.5 / 83.4 | | 61.8 / 66.0 / 47.8 | | | 98.9 / 62.7 | | 71.0 |
| | SpotLight | 59.7 / 71.3 | | 55.2 / 56.3 / 36.8 | | | 98.8 / 38.0 | | 59.4 |
| | SpotLight* | 76.3 / 83.6 | | 61.4 / **70.8** / 38.8 | | | 95.4 / 78.7 | | 72.1 |
| | DIRECTER (Ours) | **78.8** / **84.8** | | **64.4** / 70.0 / **51.7** | | | 99.1 / 86.9 | | **76.5** |

which programmatically verifies adherence to specific constraints. As the original dataset contains interleaved tasks and instructions, we follow the procedure from Venkateswaran & Contractor (2025) and rewrite the prompts using the `gpt-4o-mini` API (OpenAI, 2024) to separate them in order to simplify instruction span scaling. We report prompt-level (P. Acc) and instruction-level (I. Acc) accuracy. (2) Next, to assess long-context performance, we use *LIFBench* (Wu et al., 2024), which measures a model's instruction-following capabilities across diverse long-context scenarios. We evaluate on its sub-tasks for handling structured lists (List), and performing grounded generation from one document (OD) or multiple documents (MD), reporting the Automated Rubric-based Score (ARS). (3) Lastly, to assess reasoning under formatting constraints, we designed *GSM8K-Format*, a new benchmark based on the *GSM8K* dataset (Cobbe et al., 2021), with a formatting component inspired by Zhang et al. (2023). On this benchmark, we report both formatting (F. Acc) and task (T. Acc) accuracy. More detailed information is presented in Appendix A.2.

**Baselines.** We compare DIRECTER with three groups of baselines. *(i) Zero-shot*: Standard decoding without intervention. *(ii) Prompting Baselines*: We evaluate three strategies adapted from *PASTA* (Zhang et al., 2023): the *\*-marked* and *"-marked* baselines, which enclose instructions with symbols, and a *Few-shot* baseline that uses exemplars. *(iii) Steering Baselines*: Methods that modify internal activations at decoding time. Specifically, we include *PASTA* (Zhang et al., 2023), which suppresses attention scores on non-instruction tokens, and *SpotLight* (Venkateswaran & Contractor, 2025), which dynamically adjusts attention to maintain a target proportion on instruction tokens. For both methods, we evaluate default and tuned configurations. Details are provided in Appendix A.3.

**Implementation details.** Our main experiments use Llama-3.1-8B-Instruct (Dubey et al., 2024) except Table 2; to test generalization, we also evaluate on Llama-3.2-1B-Instruct and several Qwen-2.5-Instruct (Team, 2024) models (3B, 7B, 14B). All experiments use greedy decoding. For PASTA and SpotLight, we first used the official settings (attention scaling coefficient $\alpha = 0.01$ and target attention mass $\psi_{\text{target}} = 0.3$, respectively). Since these settings often led to oversteering, we performed a hyperparameter search for each method. We then selected the best-performing configuration on IFEval and applied this single setting to all other benchmarks for consistency. These tuned configurations, indicated with * (*e.g.*, PASTA*) each uses $\alpha = 0.1$ and $\psi_{\text{target}} = 0.1$. Further details on this process are available in the Appendix A.3. For DIRECTER, we use a fixed key scaling factor $\alpha = 100$ and plausibility threshold $\beta = 0.5$ across all tasks without any task-specific tuning.

## 4.2 MAIN RESULTS

As shown in Table 1, DIRECTER consistently outperforms all baselines across the evaluated benchmarks. Compared to the *zero-shot* baseline, DIRECTER improves the average score by 6.5% and demonstrates particularly strong gains on the strict instruction-following task, IFEval. Unlike other steering methods, which often sacrifice task correctness for instruction following and lead to oversteering, DIRECTER maintains a superior balance. This is evident on GSM8K-Format, where com-

Table 2: **Performance across model scales.** The best and second-best scores are highlighted in **bold** and underline, respectively.

| Method | Llama-3.2 | Qwen-2.5 | | |
|---|---|---|---|---|
| | **1B** | **3B** | **7B** | **14B** |
| Zero-shot | 61.3 | 63.9 | 72.4 | 81.6 |
| *-marked | 61.4 | 61.7 | 70.1 | 79.7 |
| "-marked | 56.0 | 61.5 | 70.4 | 78.5 |
| Few-shot | 56.0 | **67.8** | 71.5 | 81.9 |
| PASTA* | 59.7 | 65.2 | 73.0 | 80.1 |
| SpotLight* | 60.6 | 62.8 | **74.9** | 81.7 |
| DIRECTER (Ours) | **61.6** | 67.1 | 74.4 | **83.5** |

Table 3: **Ablation study.** The best and second-best scores are highlighted in **bold** and underline, respectively.

| Method | Accuracy |
|---|---|
| Zero-shot | 77.5 |
| DIRECTER | **81.8** |
| + Ranking reversed | 79.0 |
| + Steer random layers | 80.2±0.7 |
| + Steer random tokens | 79.2±1.1 |

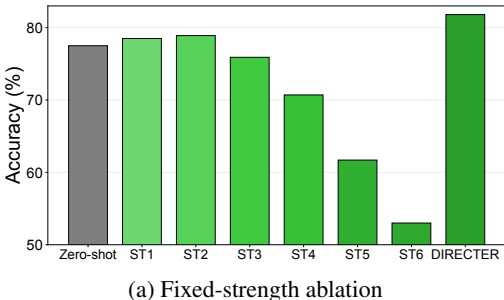

(a) Fixed-strength ablation

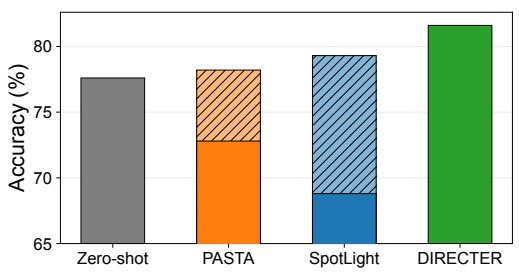

(b) Compatibility with other steering baselines

Figure 2: **Ablation studies for plausibility-guided decoding. (a)** Performance of DIRECTER compared to variants using a fixed steering strength (ST$k$, where steering is applied to $2^{k-1}$ top-ranked layers). **(b)** Applying our plausibility filter to other steering methods mitigates oversteering and improves performance. For each method, the solid bar represents the results from original version, while the hatched bar includes our filter.

peting methods suffer a significant drop in task accuracy when steered for strict formatting (*e.g.,* PASTA*: 82.7 → 62.7), while DIRECTER achieves high performance on both metrics.

To further evaluate the generalizability of DIRECTER, we assess its performance across various model families and scales on the IFEval benchmark. As detailed in Table 2, DIRECTER demonstrates robust performance gains on models ranging from 1B to 14B parameters (Llama-3.2 for 1B and Qwen-2.5 for others). This contrasts with prompting-based methods, whose effectiveness can be inconsistent across different models. For instance, the `*-marked` prompt is effective on Llama-3.1-8B-Instruct but performs poorly on Qwen models, suggesting its efficacy can be highly dependent on specific training data. Notably, prior steering baselines also exhibit similar inconsistency and, in some cases, even degrade performance compared to the zero-shot baseline. This observation further underscores that methods relying on fixed, static steering configurations are vulnerable to oversteering, reinforcing the need for adaptive control mechanisms. In contrast, DIRECTER offers a more robust, model-agnostic improvement.

### 4.3 ANALYSES

In this section, we conduct a series of analyses to provide deeper insights into the properties of DIRECTER. We mainly used Llama-3.1-8B-Instruct and the *IFEval* benchmark, reporting accuracy as the mean of prompt- and instruction-level scores.

**Effectiveness of plausibility-guided decoding.** A key aspect of DIRECTER is its ability to dynamically control the *steering strength* at each decoding step. We use ST to denote steering strength, then ST$k$ applies steering to the top $2^{k-1}$ layers (*e.g.,* ST6: steer all 32 layers; ST1: steer single layer). To assess the effect of dynamically adjusting ST via the plausibility guidance, we compare the full DIRECTER with variants that keep ST fixed throughout generation. As shown in Figure 2a, the adaptive approach of DIRECTER substantially outperforms all fixed-strength versions. While a low, fixed steering strength (*e.g.,* ST1, ST2) offers a slight improvement over the baseline, increas-

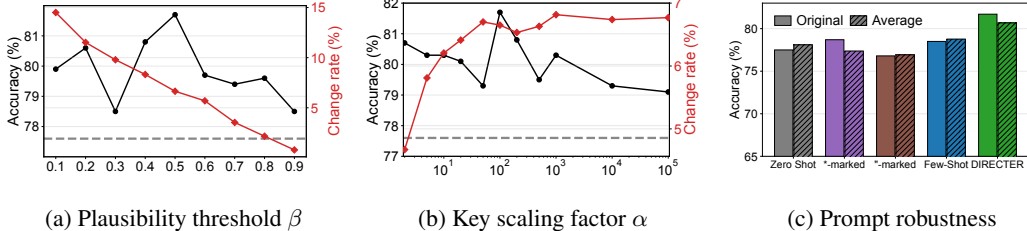

(a) Plausibility threshold $\beta$     (b) Key scaling factor $\alpha$     (c) Prompt robustness

Figure 3: **Robustness analysis of DIRECTER.** The gray dashed line indicates the baseline. Black dotted denotes accuracy (%), and red diamond denotes change rate (%). **(a)** Performance across different plausibility thresholds ($\beta$). **(b)** Stability across a wide range of scaling factors ($\alpha$). **(c)** Robustness to different prompts, showing consistent improvement on the average of four variants.

ing the strength further leads to a steep decline in performance due to oversteering. This confirms that no single, static strength is optimal across all decoding steps and that dynamically adjusting the intervention based on plausibility is critical for balancing instruction following and output quality.

To further demonstrate the plausibility guidance as a general mechanism for mitigating oversteering, we tested its compatibility with other steering baselines. In our experiments, we observed that the officially recommended settings for *PASTA* and *SpotLight* often led to severe oversteering. As shown in Figure 2b, simply integrating our plausibility filter as a safety gate substantially mitigates this issue, improving their performance by reverting to the raw prediction for implausible tokens. While this highlights the modularity and effectiveness of the plausibility check, the fully integrated approach of DIRECTER still achieves a more favorable balance of performance. (See Appendix D.5 for detailed explanation.)

**Effectiveness of the layer ranking strategy.** To validate that our attention sensitivity ranking provides an effective way to select layers for steering, we first conduct an ablation study comparing DIRECTER with variants that alter the layer selection process (Table 3). When the layer ranking order is reversed (+ *Ranking reversed*), which means that the most sensitive layers are removed first, performance degrades significantly from 81.8% to 79.0%. This confirms that our sensitivity metric correctly identifies influential layers and that removing them prematurely is detrimental. Next, when we replace our ranking with a random ordering of layers (+ *Steer random layers*), the accuracy of 80.2% is still substantially lower than our full method. This demonstrates that a principled layer selection strategy is critical for optimal performance. Additional ablations on alternative formulations of our metric are provided in Appendix D.1.

To confirm that the performance gains stem from steering the KV cache of the appropriate tokens, we also test a variant that applies scaling to randomly selected positions within the prompt instead of the identified instruction span (+ *Steer random tokens*). While this results in a notable performance drop compared to our full method, its performance still remains slightly above the zero-shot baseline. This result verifies that the gains come from amplifying the specific instructions, not arbitrary parts of the context. This suggests that our method can be safely applied even in scenarios with a partially incorrect instruction span, as it does not risk degrading performance below the baseline.

**Hyperparameter and prompt robustness.** We further analyze the effect of the plausibility threshold $\beta$ in Figure 3a. This hyperparameter controls the frequency of steering interventions; lower values of $\beta$ relax the plausibility check, leading to a higher token change rate. While optimal performance is achieved around $\beta \approx 0.5$, our method remains superior to the baseline (*i.e.*, no steering) across the full range of tested values. Additional robustness results across diverse datasets and model scales are provided in Appendix E, further confirming the stability of DIRECTER with respect to the plausibility threshold $\beta$.

DIRECTER is highly robust to the key scaling factor $\alpha$, with performance remaining stable across several orders of magnitude ($10^1 \leq \alpha \leq 10^5$) (Figure 3b). The token change rate saturates around $\alpha \approx 10^2$, after which further increases have a negligible impact on steering interventions. This supports our finding that $\alpha$ is not an effective parameter for fine-grained control of steering strength.

Finally, we assess the robustness of DIRECTER against variations in prompt design, specifically testing whether it is biased by the instruction's position or template style. We designed four variants,

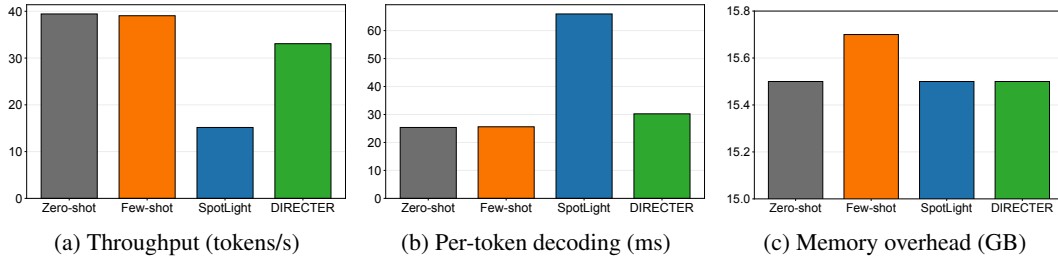

(a) Throughput (tokens/s)   (b) Per-token decoding (ms)   (c) Memory overhead (GB)

Figure 4: **Inference efficiency analysis.** DIRECTER maintains (**a**) competitive throughput and (**b**) per-token decoding speed while (**c**) adding negligible memory overhead.

each differing in prompt format and the location of the instruction. As shown in Figure 3c, the performance of DIRECTER remains stable across these variations. The figure compares the result from our main experimental prompt (solid bar) with the average performance across the four new variants (hatched bar), showing only a minimal difference. This stability confirms that the benefits of our adaptive steering are orthogonal to prompt design, establishing DIRECTER as a general-purpose enhancement module. See Appendix A.4 for the detailed prompt templates.

**Inference efficiency.** We evaluate the inference efficiency of DIRECTER on three metrics: *throughput*, *per-token decoding time*, and *memory overhead* (Figure 4) relative to key baselines.[3] Although its one-time layer ranking incurs an initial latency cost, DIRECTER maintains high overall efficiency. The worst case arises when the number of generated tokens is very small, since the fixed ranking overhead dominates the total runtime. Even in this scenario, however, DIRECTER maintains competitive throughput and remains faster than prior steering methods such as SpotLight as detailed in Appendix D.3. Overall, its throughput is only ≈16% lower than the zero-shot baseline and over 2× faster than SpotLight, with a minimal ≈20% increase in per-token decoding time. The memory overhead is also negligible, as temporary attention output is discarded immediately after the ranking process. Notably, our direct KV cache steering approach is compatible with standard optimizations like FlashAttention (Dao et al., 2022; Dao, 2023), an advantage unavailable to attention-level interventions.

**Generation quality and task fidelity.** To ensure that the improvements on structured benchmarks like IFEval do not come at the cost of performance in open-ended generation, we evaluated DIRECTER's impact on text quality and underlying task fidelity. Outputs were evaluated by `gpt-4o-mini` as an automated judge, which scored *text quality* (*e.g.*, fluency, coherence) on a 1-5 scale and assessed *task fidelity* by judging whether each output successfully met the prompt's primary goal, reporting the aggregate success rate. As shown in Table 4, DIRECTER maintains a high text quality score on par with strong non-intervention baselines, and notably higher than other steering methods like PASTA* and SpotLight*, which show a slight degradation. More importantly, DIRECTER achieves the highest task accuracy (≈ 92%), surpassing all prompting and steering baselines.

As detailed in Appendix E.3, additional human evaluation corroborates these findings, showing that DIRECTER preserves both text quality and task fidelity. While annotators rated the responses produced by PASTA as inferior in quality compared to the non-intervention baseline, DIRECTER was evaluated as comparable to, or in some cases even better than, the baseline. These results demonstrate that DIRECTER enhances instruction following without the common trade-off, simultaneously preserving generation quality while improving the model's ability to adhere to the core task.

Table 4: **LLM-based evaluation of generation quality and task fidelity.**

| Method | Text Quality | Task Fidelity (%) |
|---|---|---|
| Zero-Shot | 4.39 | 84.8 |
| *-marked | **4.43** | 87.6 |
| ''-marked | 4.37 | 84.6 |
| Few-Shot | 4.37 | 87.4 |
| PASTA* | 4.24 | 81.1 |
| SpotLight* | 4.33 | 85.7 |
| DIRECTER (Ours) | 4.40 | **91.7** |

---

[3]We exclude PASTA since it requires costly pre-computation to profile attention heads. Our analysis targets methods with overhead only at inference. See Appendix A.3 for detailed information.

Table 5: **Qualitative examples**. IFEval response comparison between the baseline and DIRECTER. More examples are provided in Appendix F.1.

| **Prompt** |
| --- |
| Write a song that critiques the song "We Are Never Ever Getting Back Together" by Taylor Swift. Wrap your entire response with double quotation marks. Do not mention the word Taylor, Swift, or Together. |

| **Response** | |
| --- | --- |
| BASELINE | "We Are Never Ever Getting Back Together" was a hit, a chart-topper's cry. A song of heartbreak, a tale of goodbye. But is it really over, or just a game? A cycle of love, a never-ending flame [...] Let's find a love that's real, and not just a game. We are never ever getting back together, no way. Let's find a love that's true, and not just a phase." |
| DIRECTER | "Verse 1: You said you're done, but I know you're not—You're just a cycle, a pattern we've been caught. You break my heart, but I still come back, a fool for love, that's just a fact [...] Chorus: We're never ever getting over this; We're stuck in a loop, it's just a mess. You say you're done, but I know you're lying—We're just a game, and I'm still trying." |

# 5 CONCLUSION

In this paper, we introduced DIRECTER, a novel inference-time activation steering method for better instruction following that addresses the limitations from oversteering and static configurations common in prior steering techniques. The core idea of DIRECTER is a plausibility-guided decoding loop that dynamically modulates steering strength by adjusting the number of steered layers, based on layer ranking with attention sensitivity. This adaptive approach, validated by extensive ablations, significantly improves instruction-following on benchmarks like IFEval while introducing only modest computational overhead. By moving beyond fixed configurations towards a step-wise, self-correcting control loop, DIRECTER demonstrates the promise of dynamic, mechanistic interventions and provides a robust framework for enhancing the reliability and controllability of LLMs.

**Limitations and future direction.** While DIRECTER achieves consistent improvements across diverse benchmarks, there remain opportunities for further refinement. Our attention sensitivity metric, though empirically effective, is not based on a formal theoretical framework; developing more principled metrics could improve performance and even enable efficient ranking strategies. Also, our evaluation assumes cleanly separable tasks and instructions similar to prior works. However, in real-world scenarios, instructions are often embedded within tasks in a single prompt, requiring complementary techniques. While we demonstrate DIRECTER remains effective in this scenario with a simple automatic span detection method (see Appendix D.2), more sophisticated method such as (Zhang et al., 2024) could be beneficial.

ETHICS STATEMENT

By enhancing the ability of LLMs to follow instructions, DIRECTER offers a practical method for improving their reliability (Askell et al., 2021) and controllability (Elhage et al., 2021) in real-world applications. This enhanced alignment with user intent is a critical step toward building safer AI systems (Bai et al., 2022b). The core principles of our approach, based on plausibility-guided dynamic steering control, are broadly applicable in domains requiring strict adherence to operational constraints.

However, advanced model control mechanisms also present potential risks. The fine-grained control offered by DIRECTER could be exploited by malicious actors to circumvent safety alignments and generate harmful or misleading content (Yao et al., 2024; Lin et al., 2025). Furthermore, because DIRECTER prioritizes following instructions over evaluating their ethical content, it could amplify the underlying biases of a model if given biased instructions (Bender et al., 2021; Anil et al., 2024). To address these concerns, we advocate for responsible deployment of DIRECTER within a comprehensive safety framework (Le Bras et al., 2020; Inan et al., 2023), which should include robust content filtering and continuous monitoring. On a practical note, while our method introduces an inference overhead, its design incorporates a gating mechanism to maintain efficiency by minimizing unnecessary computations.

REPRODUCIBILITY STATEMENT

We provide implementation details (*e.g.*, models, hyperparameters, and prompt design) and experiment setups (*e.g.*, datasets and evaluation metrics) in Section 4.1 and Appendix A. The complete source code for our implementation is available at `https://github.com/mjk0618/director`.

ACKNOWLEDGMENTS

All authors are affiliated with the Department of Artificial Intelligence at Yonsei University. This research was supported in part by Institute for Information & communications Technology Planning & Evaluation (IITP) grant funded by the Korea government (MSIT) (No. RS-2020-II201361, Artificial Intelligence Graduate School Program (Yonsei University); No. RS-2025-25442405, Development of a Self-Learning World Model-Based AGI System for Hyperspectral Imaging; No. RS-2025-02215344, Development of AI Technology with Robust and Flexible Resilience Against Risk Factors).

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

# A  EXPERIMENT DETAILS

## A.1  EXPERIMENTAL SETUP

We implemented all algorithms using `PyTorch` and the Hugging Face `transformers` (Wolf et al., 2020) library. All experiments were conducted on a single NVIDIA H100 GPU server.

## A.2  DATASET DETAILS

### A.2.1  IFEVAL

IFEval (Zhou et al., 2023) is a benchmark designed to programmatically evaluate the instruction-following capabilities of Large Language Models, containing 541 prompts across 25 diverse constraint types. It employs an automatic evaluation framework based on four key metrics. *Prompt-level* accuracy requires all constraints within a single prompt to be satisfied, while *instruction-level* accuracy assesses each constraint individually. These are further divided by matching criteria: *strict* accuracy demands an exact match with the expected output, whereas *loose* accuracy allows for minor, non-essential variations, such as ignoring case sensitivity or extra whitespace when checking for a forbidden word. In our main results, the reported prompt-level and instruction-level accuracies are the average of their respective strict and loose scores. Table 6 shows an example from the dataset.

Table 6: **An example from the IFEval dataset.** The original wide format is rearranged here for clarity, showing the key fields and their content for a single prompt.

| Field | Content |
|---|---|
| `key` | 1005 |
| `prompt` | Write a resume for a fresh high school graduate who is seeking their first job. Make sure to include at least 12 placeholder represented by square brackets, such as [address], [name]. |
| `instruction_id_list` | `detectable_content:number_placeholders` |
| `kwargs` | `num_placeholders:12` |

**Prompt rewriting for steering compatibility.** A key prerequisite for most test-time steering methods, including ours, is the clear separation between the primary task and its associated instructions. This separation is crucial for the efficient and accurate identification of the instruction span where steering should be applied. However, the original prompts in the IFEval benchmark often interleave these two components. To align the dataset with the requirements of steering-based methodologies, we adopted the preprocessing procedure from prior work Venkateswaran & Contractor (2025). We programmatically rewrote the original prompts using the `gpt-4o-mini` API (OpenAI, 2024) to explicitly disentangle the task from the instructions, guided by the prompt template shown in Figure 5. This process establishes a consistent and clearly demarcated structure, enabling the precise application of steering mechanisms. Table 7 illustrates the structural changes resulting from this process. For this benchmark, *steering was applied to the KV cache of tokens corresponding to the explicitly separated instruction span*.

**Analysis of dataset bias and prompt robustness.** To verify that our rewriting process does not inadvertently create a dataset biased towards DIRECTER, we report the performance of the zero-shot baseline using Llama-3.1-8B-Instruct on both the original and the rewritten versions of the IFEval dataset. Table 8 shows that the performance remains comparable, indicating that the rewriting process itself does not introduce a significant bias that would unfairly favor our method.

Furthermore, to assess the robustness of DIRECTER against variations in prompt design and instruction placement, we created four different prompt templates. Each template alters the phrasing or the position of the instructions relative to the main task. The four templates, filled with an example, are presented in Table 9. Except for this prompt robustness analysis, Template 1 was used as the main template for all other experiments.

```
You will be given a prompt within the <prompt> and </prompt> tags.
The prompt consists of a task or question (e.g.)  write an essay,
and one or more instructions (e.g.)  do not use any commas, highlight
sections, etc.
You must rewrite this prompt to separate the instructions from the
task and the new prompt should specify the task at the beginning.
You will also be given a list of instruction_ids that will specify the
instructions present in the prompt.
At the end of the new prompt list the instructions after the sentence
"Your response should follow the instructions below:\n"
Each instruction should be preceded by a hyphen or dash –
Make sure the new prompt is within the <new_prompt> and </new_prompt>
tags.

<prompt>
{prompt}
</prompt>

<instruction_ids>
{instruction_ids}
</instruction_ids>
```

Figure 5: **The prompt template used for rewriting IFEval samples.**

Table 7: **Examples of IFEval prompt rewriting.** The original prompts are separated into distinct *Task* and *Instruction* sections to simplify instruction span identification.

| Example 1 | |
|---|---|
| **Original** | I am planning a trip to Japan, and I would like thee to write an itinerary for my journey in a Shakespearean style. You are not allowed to use any commas in your response. |
| **Rewritten** | Your task is to write an itinerary for a trip to Japan in a Shakespearean style.
- Do not use any commas in your response. |
| **Example 2** | |
| **Original** | Write two jokes about rockets. Do not contain commas in your response. Separate the two jokes with 6 asterisk symbols: ******. |
| **Rewritten** | Write two jokes about rockets.
- Do not contain commas in your response.
- Separate the two jokes with 6 asterisk symbols: ****** |

Table 8: **Performance of the zero-shot baseline on the original and rewritten IFEval datasets.**

| Dataset Version | P. Acc. (%) | I. Acc. (%) | Mean Acc. (%) |
|---|---|---|---|
| Original IFEval | 74.4 | 82.1 | 78.3 |
| Rewritten IFEval | 73.5 | 81.5 | 77.5 |

### A.2.2 LIFBENCH

LIFBench (Wu et al., 2024) is a benchmark designed to evaluate the instruction-following capabilities of LLMs in long-context scenarios. It comprises 2,766 instructions across 11 distinct tasks,

Table 9: **Four prompt template variants used to evaluate the robustness of DIRECTER.** Each template alters the position of the instruction or the overall phrasing.

| Template ID | Example Prompt Filled with Data |
|---|---|
| **Template 1** | Your task is to write an itinerary for a trip to Japan in a Shakespearean style.
- Do not use any commas in your response. |
| **Template 2** | - Do not use any commas in your response.
Your task is to write an itinerary for a trip to Japan in a Shakespearean style. |
| **Template 3** | Given the following instructions, complete the task the user requested.
- Do not use any commas in your response.
Your task is to write an itinerary for a trip to Japan in a Shakespearean style. |
| **Template 4** | Your task is to write an itinerary for a trip to Japan in a Shakespearean style.
Complete the requested task by following the instructions below.
- Do not use any commas in your response. |

with context lengths extending up to 128k tokens. The benchmark is organized into three primary scenarios: *List*, which tests for precise indexing and selection from ordered lists; *OneDoc*, focusing on single-document information extraction and formatting; and *MultiDoc*, requiring retrieval or reasoning across multiple documents. The primary evaluation metric is the *Automated Rubric-based Score (ARS)*, where points for various rubric items are summed and normalized. This normalization results in a final score between 0 and 1, which functions analogously to an accuracy metric. Table 10 provides examples for each scenario. *Steering was applied to the KV cache of tokens within the instruction part of each prompt.*

**Subsampling for computational tractability.** The original LIFBench dataset includes samples with exceptionally long contexts, with some exceeding 100k tokens. Processing these samples during inference requires GPU memory that surpasses the capacity of our experimental hardware, making a full evaluation computationally intractable. To create a tractable yet representative subset for our experiments, we performed a two-stage subsampling process. First, we filtered out all samples exceeding a predefined string length threshold to eliminate the most resource-intensive cases. Subsequently, to preserve the benchmark's integrity and avoid introducing bias, we ensured that the proportional distribution of samples across all original subtasks was maintained. This was achieved by uniformly downsampling each subtask to match the retention rate of the most heavily filtered task. This procedure resulted in a final, balanced set of 467 samples, enabling a practical and fair evaluation within our resource constraints.

### A.2.3 GSM8K WITH FORMATTING CONSTRAINTS

The original GSM8K dataset (Cobbe et al., 2021) consists of 1.3k grade-school math word problems, designed to test the multi-step arithmetic reasoning capabilities of language models. Inspired by prior work on improving instruction following in LLMs (Zhang et al., 2023), we designed a variant of this benchmark, which we term *GSM8K-Format*, to evaluate a model's ability to perform a core reasoning task while simultaneously adhering to strict formatting constraints.

A primary motivation for this benchmark is to verify that improvements in instruction following do not come at the cost of task performance. To this end, we measure two metrics concurrently: *Format Accuracy (F. Acc)* and *Task Accuracy (T. Acc)*. Task accuracy is considered correct if the final numerical answer can be successfully parsed from the model's generated solution. Format accuracy, however, requires the entire output to strictly conform to a specified JSON structure, with no extraneous text. This dual-metric evaluation allows us to assess whether a model can follow stylistic instructions without degrading its underlying reasoning capabilities. Table 11 provides an example from our benchmark. For this benchmark, *steering was applied specifically to the instructions dictating the JSON format, excluding the Problem part containing the math word problem.*

Table 10: **Examples from the LIFBench dataset, categorized by scenario.**

| Scenario | Example |
|----------|---------|
| **List** | You're a searcher. You need to output the corresponding list elements based on the instructions and the list below.
Please follow the instructions directly without anything else. List to be retrieved:
1. 4f63efbe7f5111ef8b42581122bf941e
2. 4f6292227f5111ef8b42581122bf941e
...
8. 4f806d427f5111ef8b42581122bf941e
...
152. 4f78d6f47f5111ef8b42581122bf941e
153. 4f7ea64c7f5111ef8b42581122bf941e
Instruction: From the preceding list, choose an element at random that succeeds the item "4f806d427f5111ef8b42581122bf941e" and provide it as the output. |
| **OneDoc** | There are several different types of KEY SENTENCE in the input text, which are marked by special tags. These special tags a total of six kinds, respectively is "¡#Topic#¿", "¡@argument@¿", "¡!Transition!¿", "¡—Summary—¿", "¡*Evidence*¿", "¡-Concession-¿". Different tags represent different types of key sentence. If a sentence in the text is KEY SENTENCE, we will add a special tag with the same attribute to the beginning and end of the sentence. The head tag also contains id order information in the format ¡type-id¿. For example, the head tag with type '¡#Topic#¿' and id 1 is ¡#Topic#-1¿. Also note that when the head tag and tail tag attributes are inconsistent, this means that the sentence is a fake KEY SENTENCE. Please read the input text carefully and give the answer directly according to the instruction requirements.
Input text: ...
Instruction: Gather every instance of KEY SENTENCE classified as ¡*Evidence*¿. The output should be a Json list arranged by ids. If none are found, provide an empty array.
Output Example 1: [KEY SENTENCE1, KEY SENTENCE2, ...]
Output Example 2: [] |
| **MultiDoc** | You are a document manager. Here is a collection of documents. Each document includes information such as title, date, source, id, iD2 and specific article content (text). You need to read the documents and follow the instructions to give some information directly, without something else. Also note:
1. Some documents may be missing information such as title or source, which may affect the final output.
2. Some articles (i.e. values corresponding to the text keyword) may be duplicated.
Documents:
...
Instructions: Assign labels to documents in order using the provided list of ['11311', '22422', '33233', '44444']. ... |

Table 11: **An example from the GSM8K-Format dataset.** A formatting instruction is added to the original GSM8K problem.

| Task | Content |
|------|---------|
| **GSM8K** | Natalia sold clips to 48 of her friends in April, and then she sold half as many clips in May. How many clips did Natalia sell altogether in April and May? |
| **GSM8K-Format** | Read the given math problem and provide your answer in the following JSON format:
{{"solution": "<step-by-step solution>", "answer": "<final answer as a number only>"}}

Problem:
Natalia sold clips to 48 of her friends in April, and then she sold half as many clips in May. How many clips did Natalia sell altogether in April and May? |

### A.3 BASELINE REPLICATIONS AND IMPLEMENTATIONS

#### A.3.1 PASTA

PASTA (Zhang et al., 2023) improves instruction following via a two-stage process. In the first stage, *profiling*, an extra dataset is used to identify attention heads that are influential for a given task. During the second stage, *steering*, the attention scores of these selected heads are manipulated at inference time by scaling down the attention on non-instruction tokens.

The profiling stage is computationally intensive, requiring $N \times L \times H$ forward passes, where $N$ is the number of profiling examples, $L$ is the number of layers, and $H$ is the number of attention heads. For instance, profiling Llama-3.1-8B-Instruct with 1000 examples as suggested in the original work would require $1000 \times 32 \times 32$, over a million, forward passes. To ensure a fair comparison with methods that operate solely at inference time, we adapted PASTA to a few-shot profiling setting. While the original work uses up to 1000 examples per task, we used 10 examples for IFEval and GSM8K-Format, and 11 for LIFBench (one for each of its subtasks). Since IFEval lacks a public training set, we manually crafted these few-shot examples, as shown in Table 12.

Following the profiling stage, we adopted the *task-specific* head selection strategy proposed in the paper, steering a total of 50 heads. We initially used the recommended attention scaling factor of $\alpha = 0.01$, but observed performance degradation on IFEval. Consequently, we conducted a brief hyperparameter search and found $\alpha = 0.1$ to be more effective for our tasks. The results of this search are detailed in Figure 6a.

Table 12: **An example of a manually crafted IFEval benchmark.**

| Field | Content |
|---|---|
| `key` | 9000 |
| `prompt` | Draft a timeline of the Apollo 11 mission from launch to splashdown. 
 - Include exactly 10 placeholders represented by square brackets, such as [time] or [location]. 
 - Keep the response under 120 words. |
| `instruction_id_list` | `detectable_content:number_placeholders` 
 `length_constraints:number_words` |
| `kwargs` | `num_placeholders:10` 
 `relation: num_words, less_than: 121` |

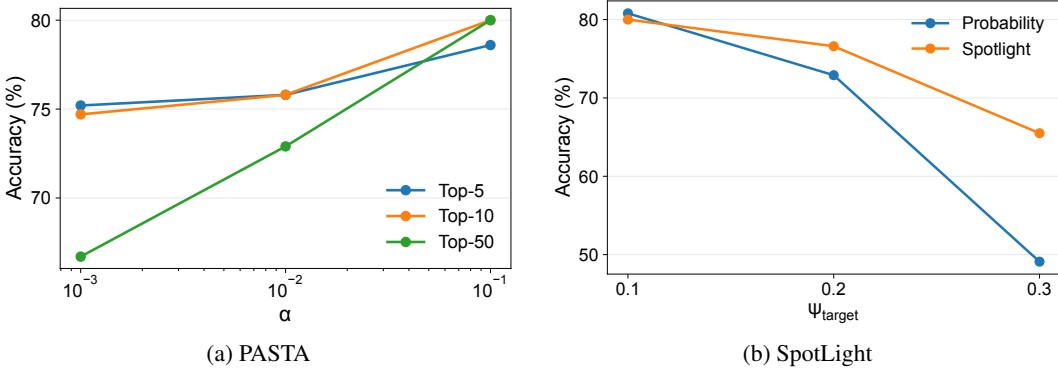

(a) PASTA

(b) SpotLight

Figure 6: **Results of the hyperparameter search on the IFEval dataset.** (**a**) PASTA (varying the scaling factor $\alpha$) and (**b**) SpotLight (varying the target attention mass $\psi_{target}$)

### A.3.2 SPOTLIGHT

As the official implementation for SpotLight (Venkateswaran & Contractor, 2025) was not publicly available, we re-implemented the method based on the description provided in the original paper. To verify our implementation, we compared the paper's proposed method (*SpotLight*), which uses an additive bias to approximate a target attention proportion, against a stricter variant that directly normalizes scores to always maintain this target (*Probability*). Figure 7 illustrates the layer-wise behavior of both versions. For all our main experiments, we used the official additive bias approach as proposed in the paper.

The original work recommends a target attention mass of $\psi_{target} = 0.3$. However, this setting led to a significant performance degradation in our experiments. Similar to our process for PASTA, we performed a hyperparameter search on IFEval (as shown in Figure 6b). Based on these results, we selected $\psi_{target} = 0.1$ for all reported scores.

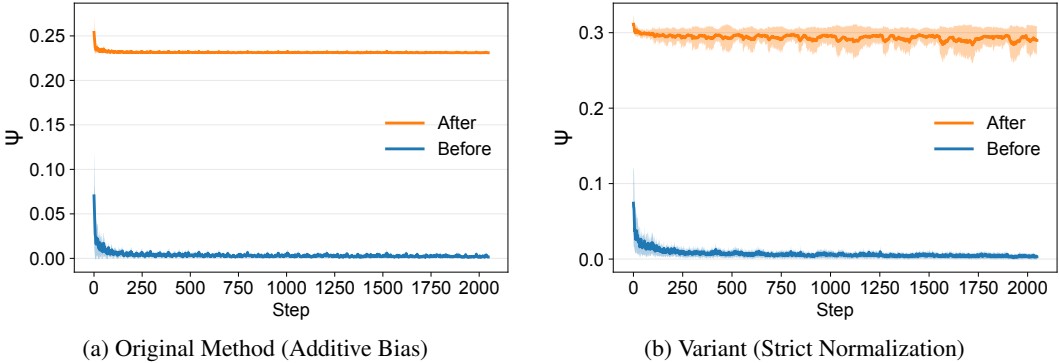

(a) Original Method (Additive Bias)  (b) Variant (Strict Normalization)

Figure 7: **Verification of our SpotLight implementation.** We compare the layer-wise attention proportion on instruction tokens for (**a**) the original method from the paper and (**b**) a variant that strictly enforces the target proportion.

### A.4 PROMPT TEMPLATES

As detailed in Section 4.3, we conducted an LLM-based evaluation to ensure that enhancing instruction-following performance does not negatively impact text quality or task fidelity on open-ended generation tasks. The prompt template provided to the LLM judge for this evaluation is shown in Figure 8.

## B ADDITIONAL RELATED WORK

This section provides a more detailed overview of the two main categories of activation steering methods discussed in Section 2 : hidden state-level and attention-level interventions. We focus on the inherent limitations of each approach that motivate our work, DIRECTER.

**Hidden state-level steering.** The primary limitation of methods that operate on hidden states is their fundamental reliance on pre-computed signals or external datasets. This category includes several approaches that, despite being adaptive in form, are restricted to pre-defined concepts: *CAST* (Lee et al., 2024) gates steering by detected semantics; *ACT* (Wang et al., 2025) adapts strength for truthfulness; and *SADI* (Wang et al., 2024a) scales influential components from contrastive signals. Consequently, even though these methods are often described as *dynamic* or *adaptive*, their steering capabilities are fundamentally restricted to these pre-defined traits. This reliance on pre-computation means the steering vector represents a static *concept space* defined offline. Such a fixed representation makes it difficult for the model to adapt to the novel, step-by-step dynamics of a specific generation process, where the ideal steering direction may shift based on the evolving context. This highlights the need for a truly online control mechanism that can respond to the immediate state of the generation.

```
You are an impartial evaluator of instruction-following tasks.

You will receive a "query" and a "response".  In this dataset,
each query contains:
(1) the task to be completed, and
(2) additional formatting instructions.

For this evaluation, IGNORE the formatting instructions entirely
because they are evaluated elsewhere.  Focus only on whether the
response completed the TASK content correctly and whether the
writing quality is adequate.

Scoring rules:
- task_fidelity:  1 if (and only if) the response fulfills the
task asked in the query; else 0.
- text_quality:  an integer from 1 to 5 reflecting clarity,
coherence, fluency, and appropriateness of tone for the task
(1=very poor, 5=excellent).

Provide also a brief justification (1-3 concise sentences) that
explains your decision, referring only to task fulfillment and
writing quality.

Return STRICT JSON with the keys:  task_fidelity, text_quality,
justification.

Query:
{query}

Response:
{response}
```

Figure 8: **The prompt template used for LLM-based evaluation.**

**Attention-level steering.** The attention mechanism has long been a focal point for interpretability research (Jain & Wallace, 2019; Wiegreffe & Pinter, 2019; Elhage et al., 2021), but its direct application to inference time steering remains a relatively under-explored area. As noted in Section 2, existing attempts in this space have notable limitations. Examples include *PASTA* (Zhang et al., 2023), which profiles beneficial heads and suppresses attention to non-instruction tokens, and *SpotLight* (Venkateswaran & Contractor, 2025), which maintains a target proportion of attention on instruction tokens. However, PASTA's profiling requires costly pre-computation on a large validation set, and SpotLight introduces relatively high latency by doubling softmax operations. More broadly, both rely on static or manually tuned configurations that do not adapt to the evolving context of the decoding process, leaving a gap for more robust, online control mechanisms like ours.

## C    METHOD DETAILS

### C.1    FULL ALGORITHM OF DIRECTER

Algorithm 2 provides a detailed walkthrough of the DIRECTER procedure. The process is divided into two main phases. *Phase 1* details the one-time layer ranking based on attention sensitivity, which is performed once after the initial prompt prefill. This phase calculates the influence of each layer and produces a static ranked list, $\mathcal{L}_{\text{ranked}}$. *Phase 2* describes the per-token generation loop, which uses this ranked list to perform plausibility-guided decoding (Section 3.2), adaptively moderating steering strength at each step.

---

**Algorithm 2:** Detailed Procedure of DIRECTER

---

**Input:** Prompt $\mathbf{x}$, scaling factor $\alpha$, plausibility threshold $\beta$
**Output:** Generated sequence $\mathbf{y}$
`// Phase 1:  Prefill and Layer Ranking`
1   $\mathcal{C} \leftarrow \text{Prefill}(\mathbf{x})$
2   Initialize layerwise Attention Sensitivity $S \leftarrow \text{zeros}(L)$
3   **for** $\ell = 1$ **to** $L$ **do**
4      $\mathcal{C}_{(\ell)} \leftarrow \text{ScaleCache}(\mathcal{C}, \{\ell\}, \alpha)$
         `// Steer key cache of instruction tokens of only layer ℓ`
5      $\{\mathbf{H}_{\text{pre}}^{(j)}\}_{j=1}^{L}, \{\mathbf{H}_{\text{post}}^{(j)}\}_{j=1}^{L} \leftarrow \text{GetHiddenStates}(\mathcal{C})$ `// Get baseline hidden states`
6      $\{\mathbf{H}_{\text{pre}}^{(j,\ell)}\}_{j=1}^{L}, \{\mathbf{H}_{\text{post}}^{(j,\ell)}\}_{j=1}^{L} \leftarrow \text{GetHiddenStates}(\mathcal{C}_{(\ell)})$ `// Get steered hidden states`
7      **for** $j = 1$ **to** $L$ **do**
8          $D_j(\ell) \leftarrow \Big(\text{dist}(\mathbf{H}_{\text{pre}}^{(j)}, \mathbf{H}_{\text{post}}^{(j,\ell)}) - \text{dist}(\mathbf{H}_{\text{pre}}^{(j)}, \mathbf{H}_{\text{post}}^{(j)})\Big) + \Big(\text{dist}(\mathbf{H}_{\text{pre}}^{(j,\ell)}, \mathbf{H}_{\text{post}}^{(j)}) - \text{dist}(\mathbf{H}_{\text{pre}}^{(j)}, \mathbf{H}_{\text{post}}^{(j)})\Big)$
9          $S[\ell] \leftarrow S[\ell] + D_j(\ell)$
10   $\mathcal{L}_{\text{ranked}} \leftarrow \text{Argsort}(S, \text{descending})$
11   $\mathbf{y} \leftarrow$ tokens from $\mathbf{x}$

`// Phase 2:  Plausibility-Guided Decoding`
12   **while** *not end-of-sequence* **do**
13      $p_t \leftarrow \text{Forward}(\mathcal{C})$ `// Get raw probability distribution`
14      $\mathcal{L}_{\text{cand},t} \leftarrow \mathcal{L}_{\text{ranked}}$ `// Initialize candidate layers for this step`
15      $\text{accepted} \leftarrow \text{False}$
16      **while** $|\mathcal{L}_{cand,t}| > 0$ **and not** *accepted* **do**
17          $\mathcal{C}_{\text{scaled}} \leftarrow \text{ScaleCache}(\mathcal{C}, \mathcal{L}_{\text{cand},t}, \alpha)$
18          $\widetilde{p}_t \leftarrow \text{Forward}(\mathcal{C}_{\text{scaled}})$ `// Get steered distribution`
19          $i_t^* \leftarrow \text{argmax}(p_t); \quad \widetilde{i}_t^* \leftarrow \text{argmax}(\widetilde{p}_t)$
         `// Plausibility check`
20          **if** $p_{t,\widetilde{i}_t^*} \geq \beta \cdot p_{t,i_t^*}$ **then**
21              $p_{\text{final}} \leftarrow \widetilde{p}_t$
22              $\text{accepted} \leftarrow \text{True}$
23          **else**
24              $k \leftarrow \lfloor |\mathcal{L}_{\text{cand},t}|/2 \rfloor$
25              $\mathcal{L}_{\text{cand},t} \leftarrow \mathcal{L}_{\text{cand},t}[:k]$ `// Reduce strength by halving layers`
26      **if** *not accepted* **then**
27          $p_{\text{final}} \leftarrow p_t$ `// Fallback to original if no version is plausible`
28      $t_{\text{next}} \leftarrow \text{Sample}(p_{\text{final}})$
29      $\mathcal{C} \leftarrow \text{UpdateCache}(\mathcal{C}, t_{\text{next}})$
30      Append $t_{\text{next}}$ to $\mathbf{y}$
31   **return** $\mathbf{y}$

---

## C.2   SIMPLIFICATION OF THE ATTENTION SENSITIVITY METRIC

As formulated in Section 3.2, the attention sensitivity score is conceptually derived from the total disturbance a perturbation at layer $\ell$ causes relative to a baseline. The full expression is:

$$\text{Sensitivity}(\ell) = \frac{1}{L} \sum_{j=1}^{L} \Bigg[ \Big(\text{dist}(\mathbf{H}_{\text{pre}}^{(j)}, \mathbf{H}_{\text{post}}^{(j,\ell)}) - \text{dist}(\mathbf{H}_{\text{pre}}^{(j)}, \mathbf{H}_{\text{post}}^{(j)})\Big)$$
$$+ \Big(\text{dist}(\mathbf{H}_{\text{pre}}^{(j,\ell)}, \mathbf{H}_{\text{post}}^{(j)}) - \text{dist}(\mathbf{H}_{\text{pre}}^{(j)}, \mathbf{H}_{\text{post}}^{(j)})\Big) \Bigg] \tag{4}$$

This can be rearranged to separate the terms dependent on the perturbed layer $\ell$:

$$\text{Sensitivity}(\ell) = \frac{1}{L} \sum_{j=1}^{L} \Big(\text{dist}(\mathbf{H}_{\text{pre}}^{(j)}, \mathbf{H}_{\text{post}}^{(j,\ell)}) + \text{dist}(\mathbf{H}_{\text{pre}}^{(j,\ell)}, \mathbf{H}_{\text{post}}^{(j)})\Big) - C \tag{5}$$

where the term $C = \frac{2}{L} \sum_{j=1}^{L} \text{dist}(\mathbf{H}_{\text{pre}}^{(j)}, \mathbf{H}_{\text{post}}^{(j)})$ is a constant with respect to $\ell$, as it is calculated from the single, unperturbed forward pass. Since layer ranking is an ordinal operation, subtracting this

common constant from all scores does not alter the final rank order. For computational efficiency, we therefore use the simplified metric for the ranking procedure:

$$\text{Sensitivity}_{\text{simplified}}(\ell) \propto \sum_{j=1}^{L} \left( \text{dist}(\mathbf{H}_{\text{pre}}^{(j)}, \mathbf{H}_{\text{post}}^{(j,\ell)}) + \text{dist}(\mathbf{H}_{\text{pre}}^{(j,\ell)}, \mathbf{H}_{\text{post}}^{(j)}) \right) \tag{6}$$

A potential concern with this metric is that one might assume a linear relationship between a layer's index and its sensitivity ranking. This assumption stems from the idea that a layer's sensitivity might correlate with its depth, given that its exposure to propagated effects from steering interventions varies by position. However, contrary to this intuition, the relationship is not linear. As we demonstrate in Appendix D.4, the actual distribution of the most influential layers changes depending on the task. We hypothesize that this is attributable to the complex non-linear interactions throughout the model.

### C.3 JUSTIFICATION AND VALIDATION OF THE SENSITIVITY METRIC

In this section, we provide additional intuition regarding the design of our attention sensitivity metric (Eq. 2 and Eq. 3) and present supplementary experiments to validate our design choices.

**Rationale for summing distributional shifts.** The primary objective of our attention sensitivity analysis is to derive a principled, data-free ranking of layers based on the steering effect they provide. While estimating the precise downstream effect of steering multiple layers simultaneously is inherently challenging, we approximate this influence by isolating the marginal contribution of each layer. However, measuring only the local change at the steered layer is insufficient; a steering intervention modifies the output of the target layer, which subsequently alters the input processing of all following layers. To address this, our metric captures two distinct effects: the *direct impact* on the steered layer's attention output, and the *propagated impact* on the attention input of subsequent layers. By summing these distributional shifts across all layers, we capture the aggregate impact caused by the intervention. This summation serves as a proxy for the layer's global influence, aggregating step-wise shifts to estimate how effectively a steering applied at a specific layer propagates through the model to alter the final output distribution.

**Preference for cosine distance over norm-based metrics.** To quantify these shifts, we select cosine distance as the metric of choice due to the underlying mechanics of our steering method. We apply scaling to the key vectors of the specified instruction span, which directly alters the attention scores. Given that the attention mechanism concludes with a Softmax operation, absolute magnitude changes are largely renormalized; the intervention therefore manifests primarily as a redistribution of attention weights. This redistribution, in turn, changes the coefficients used for the weighted averaging of value vectors. Consequently, norm-based metrics, such as the L2 norm, would be susceptible to magnitude fluctuations that do not necessarily correlate with directional shifts in the representation. In contrast, cosine distance explicitly captures this directional shift, serving as a robust proxy for the semantic change in the attention output. This design choice is further supported by methodologies in KV cache compression literature, which leverage cosine similarity to identify redundant layers where input-output representations remain stable (Liu et al., 2023b; Wang et al., 2024b). We apply the inverse logic: seeking layers where a steering intervention produces the maximum directional shift, thereby signaling high steering leverage.

**Empirical validation.** To further validate the specific choice of cosine distance, we conducted an additional ablation study comparing our cosine-based ranking against an L2-norm-based ranking. As shown in Table 13, ranking by cosine distance consistently yields higher accuracy across diverse benchmarks compared to L2. This confirms that directional sensitivity is a more accurate predictor of steering efficacy than magnitude-based sensitivity in this context.

It is also worth noting that while sensitivity-based ranking yields the optimal results, the plausibility-guided decoding loop remains highly effective even with simpler layer selection strategies. The ranking procedure acts as a modular component that enhances efficiency and precision, allowing it to be further improved or replaced in future work.

Table 13: **Comparison of ranking metrics:** Cosine Distance vs. L2 Norm.

| Method | IFEval | GSM8K-Format | LIFBench |
|---|---|---|---|
| | Mean Acc. | F. Acc. / T. Acc. | Avg. |
| Baseline | 77.5 | 79.2 / 82.7 | 57.6 |
| L2-based Ranking | 80.5 | 98.5 / 85.1 | 58.6 |
| **DIRECTER (Cosine-based)** | **81.8** | **99.1 / 86.9** | **62.0** |

## C.4 CONTROL OF STEERING STRENGTH

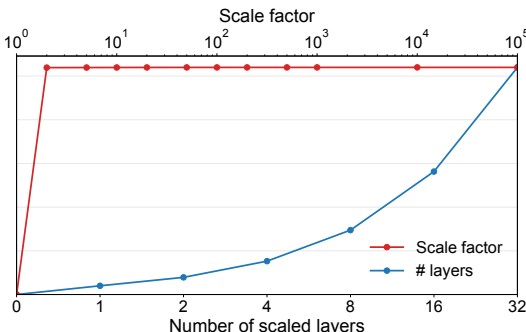

Figure 9: **Impact of steering parameters on representational shift.** Varying the scaling factor $\alpha$ (red) has a saturating effect, while varying the number of steered layers (blue) provides fine-grained control.

An important design choice in DIRECTER is to modulate the steering strength by adjusting the number of scaled layers rather than the key scaling factor, $\alpha$. To justify this, we analyze how each parameter influences the overall representational shift within the model, which we use as a proxy for the intensity of the steering intervention. As shown in Figure 9, the two parameters offer vastly different levels of control. When varying the scaling factor $\alpha$ (red curve), we observe that the steering effect saturates almost instantly, acting as a binary on/off switch rather than a gradual control knob. In contrast, when we progressively increase the number of top-ranked layers being steered (blue curve), the total shift increases smoothly and monotonically. This demonstrates that modulating the number of steered layers provides a more effective and stable mechanism for the adaptive control loop in DIRECTER.

## C.5 KEY SCALING VS. VALUE SCALING

As discussed in Section 3.1, KV cache steering can be applied by scaling either keys or values while we exclusively focus on *key scaling* in this work. As demonstrated previously in Figure 3b, our method is highly robust to the choice of the key scaling factor, consistently outperforming the zero-shot baseline across a wide range of values. It is true that value scaling also yields improvements over the baseline, as our plausibility guidance mechanism effectively mitigates oversteering. Nevertheless, its performance is suboptimal compared to key scaling, a trend clearly illustrated in Figure 10. This empirical result, combined with our goal of minimizing hyperparameters, solidified our decision to focus on a single, more effective scaling target. We hypothesize that this performance discrepancy arises from the distinct functional roles of queries, keys, and values within the attention mechanism. A deeper investigation into this interaction is beyond the scope of this paper and remains an avenue for future research.

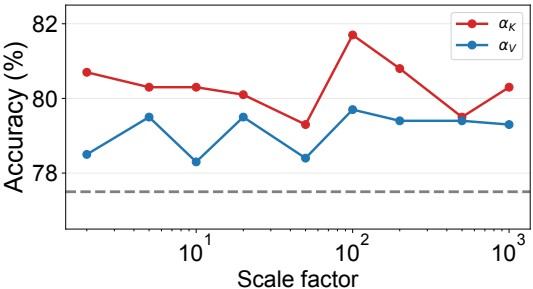

Figure 10: **Comparison between key scaling ($\alpha_K$) and value scaling ($\alpha_V$).** Key scaling consistently achieves higher performance than value scaling across various scale factors.

## C.6    PLAUSIBILITY CRITERION

We considered alternative methods for the plausibility constraint beyond the probability ratio gate in Eq. 1, namely acceptance tests based on KL and JS divergences. On a 20% subsample of IFEval, we changed only the acceptance test while keeping all other settings identical, tuned thresholds on a held-out split, and evaluated three independent runs on a disjoint subset. As summarized in Table 14, the probability ratio-based method achieves the best accuracy, whereas KL and JS divergence-based criteria generally underperform and show greater sensitivity to threshold choice.

| Criterion | Threshold | Accuracy (%) |
|---|---|---|
| Baseline (Zero-shot) | – | $72.6 \pm 1.99$ |
| Probability ratio ($\beta$) | 0.3 | $75.7 \pm 0.53$ |
| Probability ratio ($\beta$) | 0.5 | $\mathbf{78.2 \pm 0.82}$ |
| Probability ratio ($\beta$) | 0.7 | $77.7 \pm 2.11$ |
| JS divergence ($\tau_{\mathrm{JS}}$) | 0.05 | $74.8 \pm 0.85$ |
| JS divergence ($\tau_{\mathrm{JS}}$) | 0.10 | $72.6 \pm 2.15$ |
| JS divergence ($\tau_{\mathrm{JS}}$) | 0.15 | $74.6 \pm 2.94$ |
| JS divergence ($\tau_{\mathrm{JS}}$) | 0.20 | $73.3 \pm 2.40$ |
| KL divergence ($\tau_{\mathrm{KL}}$) | 0.1 | $72.8 \pm 0.46$ |
| KL divergence ($\tau_{\mathrm{KL}}$) | 0.2 | $74.3 \pm 1.20$ |
| KL divergence ($\tau_{\mathrm{KL}}$) | 0.3 | $74.3 \pm 2.01$ |
| KL divergence ($\tau_{\mathrm{KL}}$) | 0.4 | $71.6 \pm 2.25$ |

Table 14: **Comparison of plausibility criteria on IFEval subset.** Accuracy is reported as mean $\pm$ standard deviation over three runs.

We attribute this empirical gap to the characteristics of KL and JS divergence. These measures aggregate shifts over the entire vocabulary, so diffuse changes in the long tail region can inflate divergence even when the top-1 token remains acceptable, resulting in unnecessary rejections. Moreover, KL divergence has no fixed global scale, while JS divergence is bounded but still context dependent, making it difficult to select a single stable threshold. Both metrics require computing the steered distribution, which prevents compatibility with our top-2 probability-based skipping mechanism that avoids unnecessary forward passes. Finally, our design aligns with previous contrastive decoding frameworks (Li et al., 2022; Chuang et al., 2023; Xu et al., 2024) that compare top-token preferences across distributions, and it integrates naturally with our plausibility-guided loop.

## D  ADDITIONAL ANALYSES

### D.1  ABLATION STUDY ON THE ATTENTION SENSITIVITY METRIC

While the main experiments demonstrate the effectiveness of the attention sensitivity-based layer ranking in DIRECTER, the formulation of our metric is guided by the intuition that an effective sensitivity measure should capture both the direct and propagated effects of a perturbation. To provide a stronger empirical foundation for this design choice, we conducted an analysis of several alternative formulations for the sensitivity metric.

As defined in Equation 2, our proposed metric is composed of two key terms that measure the disturbance caused by a perturbation at layer $\ell$: (1) the direct impact on a layer $j$'s attention output, and (2) the propagated impact on layer $j$'s attention input. We experimented with several metric variants, including those that rely on only one of these two components. Our findings indicate that the complete metric, which incorporates both terms, consistently outperforms these simplified alternatives. This result provides strong empirical support for our design, suggesting that considering both the direct and propagated effects is crucial for accurately identifying the most influential layers for steering.

Table 15: **Performance of different sensitivity metric variants.** $d(\cdot, \cdot)$ denotes the cosine distance.

| Equation | P. Acc (%) | I. Acc (%) | Mean Acc (%) |
|---|---|---|---|
| $d(\mathbf{H}_{\text{post}}^{(j)}, \mathbf{H}_{\text{post}}^{(j,\ell)})$ | 75.8 | 83.0 | 79.4 |
| $d(\mathbf{H}_{\text{post}}^{(j)}, \mathbf{H}_{\text{post}}^{(j,\ell)}) - d(\mathbf{H}_{\text{pre}}^{(j)}, \mathbf{H}_{\text{pre}}^{(j,\ell)})$ | 76.9 | 83.4 | 80.2 |
| $d(\mathbf{H}_{\text{pre}}^{(j)}, \mathbf{H}_{\text{post}}^{(j,\ell)})$ | 78.5 | 84.5 | 81.5 |
| $d(\mathbf{H}_{\text{post}}^{(j)}, \mathbf{H}_{\text{pre}}^{(j,\ell)})$ | 78.5 | 84.5 | 81.5 |
| DIRECTER | 78.8 | 84.8 | 81.8 |

### D.2  AUTOMATIC INSTRUCTION SPAN IDENTIFICATION

One practical consideration is that our method assumes that the task description and instructions, such as formatting constraints, are explicitly separable. However, in real-world deployment scenarios, user prompts are often structured in an interleaved manner where tasks and instructions are mixed. While our primary experiments presuppose separable instructions, consistent with prior work (Zhou et al., 2023; Venkateswaran & Contractor, 2025), we also investigated the method's applicability in these more practical, interleaved settings.

Instead of relying on external APIs or larger models for prompt rewriting, we prompted the LLM (Llama-3.1-8B-Instruct) to self-extract the instruction span before beginning its main task. The steering was then applied only to the token indices corresponding to this extracted span. The results of this experiment are presented in Table 16. The specific prompt used for the instruction span extraction is shown in Figure 11.

The results demonstrate that even when using a simple self-extraction prompt for practical, interleaved instructions, DIRECTER effectively outperforms all baselines. However, this approach can fail in ambiguous cases (Table 17), suggesting that overall performance could be further improved by integrating more sophisticated instruction span detection methods.

### D.3  INFERENCE OVERHEAD ANALYSIS

We analyze the inference overhead of DIRECTER in terms of latency and memory usage, comparing it against the zero-shot, few-shot, and SpotLight baselines.

Table 16: **Performance of automatic instruction span identification on the IFEval benchmark.**

| Method | P. Acc (%) | I. Acc (%) | Mean Acc (%) |
|---|---|---|---|
| Zero-shot | 74.4 | 82.1 | 78.3 |
| ”-marked | 71.7 | 80.1 | 75.9 |
| *-marked | 75.5 | 82.9 | 79.2 |
| Few-shot | 71.0 | 79.5 | 75.2 |
| DIRECTER | **76.6** | **83.6** | **80.1** |

```
Extract exactly one substring from QUERY that expresses the
instruction to the assistant (requirements, constraints,
formatting rules, steps, or prohibitions).  Copy it verbatim
from QUERY. Return the substring ONLY --- no quotes, no labels,
no bullets, no code fences, no extra words.

EXAMPLE

QUERY:
{example_query}

OUTPUT (substring only):
{example_output}

Now do the same for the following QUERY. Output the substring
only, with nothing else:

QUERY:
{query}
```

Figure 11: **The prompt template used to instruct the LLM to self-extract the instruction span from an interleaved user query.**

**Time to first token (TTFT).** Our method, DIRECTER, requires a one-time layer sensitivity analysis after the initial prompt prefill to rank the layers. As a result, the time required to generate the first token (TTFT) is inherently longer compared to other methods that do not require this precomputation step. Table 18 shows that this initial overhead is the primary latency cost of our method.

**Decoding throughput.** While the initial TTFT is higher, this overhead is amortized over the entire generation sequence. Figure 12 provides a more detailed breakdown of the decoding throughput. As shown in Figure 12a, which visualizes the average throughput across performance percentiles, the throughput of DIRECTER in worst-case scenarios (i.e., lower percentiles) is naturally lower than its average, yet it consistently outperforms the attention-level intervention method, SpotLight.

Furthermore, Figure 12b illustrates the relationship between throughput and the total length of the generated sequence. The throughput is lower for shorter sequences because the initial overhead from our one-time layer ranking constitutes a larger proportion of the total generation time. This observation is consistent with our earlier TTFT analysis.

**Memory overhead.** A precise comparison of memory overhead is challenging, as the average length of the generated response varies across different methods (Table 19a). Nevertheless, by measuring the change in reserved memory, we can analyze the additional overhead. Table 19b shows that the memory overhead of DIRECTER is modest.

The overhead in our method stems from the need to extract attention states from both the raw and steered forward passes during the one-time layer ranking stage. We hypothesize that our current implementation could be further optimized; the `output_attentions` flag is currently enabled throughout the entire decoding process. By disabling this flag after the ranking is complete, the

Table 17: **An example of a failure case in the self-extraction of an instruction span.**

| Component | Text |
|---|---|
| Full User Prompt | Write a letter to a friend in all lowercase letters ask them to go and vote. |
| Correct Span (Expected) | in all lowercase letters |
| Incorrect Span (Extracted) | ask them to go and vote |

Table 18: **Comparison of Time to First Token (TTFT) across methods.**

| Method | Mean TTFT (s) |
|---|---|
| Zero-shot | 0.028 |
| Few-shot | 0.031 |
| SpotLight | 0.038 |
| DIRECTER (Ours) | **1.160** |

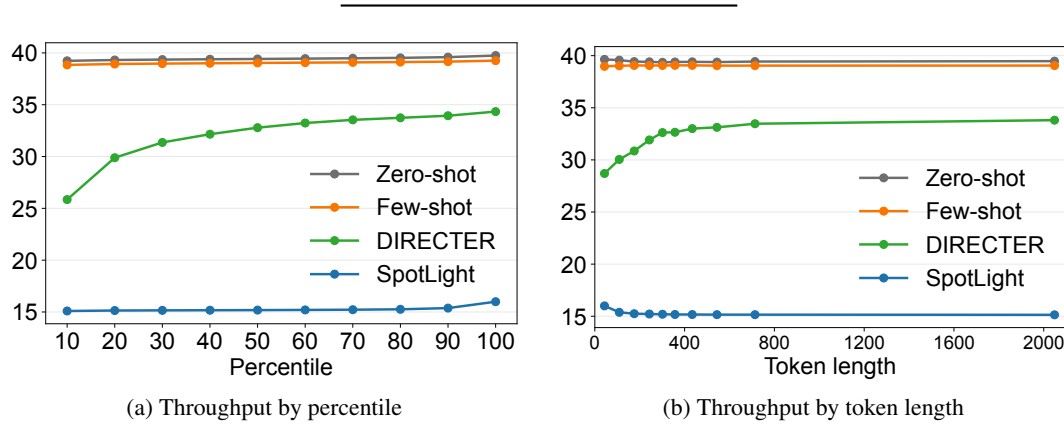

(a) Throughput by percentile      (b) Throughput by token length

Figure 12: **Detailed analysis of throughput (TPS). (a)** shows the average TPS across different performance percentiles. **(b)** shows the average TPS as a function of the generated token length.

average memory usage would likely approach that of the zero-shot baseline unlike attention-level steering methods that constantly require these attention outputs.

To further investigate the source of inference latency, we analyze the number of decoding trials at each generation step. Here, a *decoding trial* refers to each attempt to apply steering; the $t$-th trial in our setup involves steering a progressively reduced set of $2^{6-t}$ layers (*i.e.*, 32 layers in the first trial). Figure 13 plots a heatmap where the color intensity corresponds to the token change rate after steering.

The key observation is that when the model's initial confidence is high (a high raw token probability), the token change rate remains low even after multiple decoding trials. This indicates that steering is largely ineffective when the model is already highly confident in its prediction, and the additional forward passes for each trial do not lead to a new token being selected. This observation suggests that there remains a clear opportunity to further optimize inference latency.

## D.4 LAYER RANKING DISTRIBUTION

In this section, we analyze the distribution of layer sensitivity rankings across the different benchmarks to understand how task characteristics influence which layers are most critical for steering. Figure 14 visualizes the detailed probability distribution of each layer's sensitivity rank, calculated once at prefill. To provide a more aggregated and intuitive view of this same data, Figure 15 shows the proportion of times each layer is included as a steering candidate at various steering strength levels. In this visualization, a darker color at a lower steering strength value (higher on the y-axis)

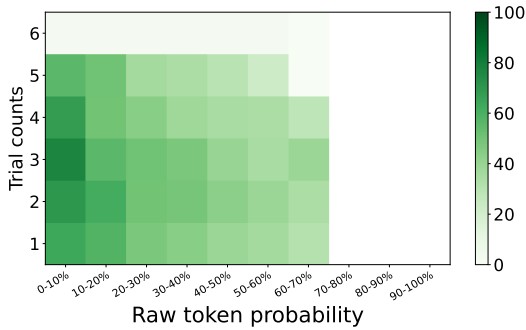

Figure 13: **Analysis of decoding trials.** The heatmap color corresponds to the token change rate.

Table 19: **Inference overhead analysis.** (**a**) shows the average number of generated tokens per response, (**b**) shows the reserved memory usage before and after generation.

<table>
<tr><td colspan="2">(a) Average generated tokens.</td><td colspan="4">(b) Average memory usage (MB).</td></tr>
</table>

| Method | Tokens | | Method | Before | After | Delta |
|---|---|---|---|---|---|---|
| Zero-shot | 387.1 | | Zero-shot | 15452 | 15546 | 94 |
| Few-shot | 329.4 | | Few-shot | 15642 | 15742 | 100 |
| SpotLight | 406.5 | | SpotLight | 15353 | 15494 | 141 |
| DIRECTER (Ours) | 387.6 | | DIRECTER (Ours) | 15351 | 15536 | 185 |

indicates that a layer is more consistently ranked among the most sensitive, as only top-ranked layers are included in low-strength candidate sets.

We observe from these distributions that the patterns vary notably across tasks. The general instruction-following benchmarks, IFEval and LIFBench, exhibit broadly similar trends, with a higher concentration of sensitivity in the early-to-middle layers. In contrast, GSM8K-Format, which combines arithmetic reasoning with formatting constraints, displays a markedly different and sparser distribution. For this task, sensitivity is concentrated in both the initial and final layers of the network, with a particularly high steering priority placed on the late layers. These distinct patterns underscore that the layers most crucial for steering are highly task-dependent, motivating our adaptive, rank-based approach over a fixed strategy.

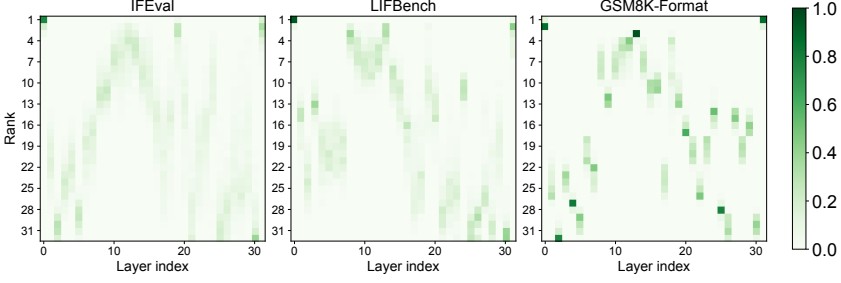

Figure 14: **The ranking distribution of each layer's sensitivity across each benchmark.**

### D.5 COMPATIBILITY OF PLAUSIBILITY-GUIDED DECODING LOOP

Our previous results have confirmed that the plausibility-guided decoding loop can be seamlessly integrated with existing steering methods to effectively control oversteering (Figure 2b). In this section, we describe the specific approach for integrating our loop with several prominent methods and present quantitative results that validate its broad compatibility and effectiveness.

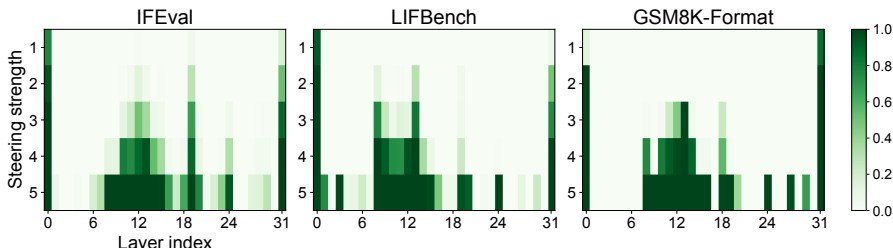

Figure 15: **An aggregated view of layer sensitivity.** This heatmap shows the proportion of times each layer is included as a steering candidate at various steering strength levels.

For PASTA, we integrated our plausibility check by modulating its attention scaling factor. We begin with a strong steering effect by setting the factor to 0.01, as recommended in the original work. If the plausibility check fails, we progressively weaken the steering by increasing the scaling factor tenfold at each step (*i.e.*, from 0.01 to 0.1, and finally to 1.0). A final scaling factor of 1.0 effectively removes the intervention, causing the model to default to its original output. For SpotLight our loop adjusts the target attention proportion, $\psi_{\text{target}}$. We start with $\psi_{\text{target}} = 0.3$, following the original work. Upon a plausibility check failure, our method gradually reduces this target by 0.1 per step. If all trials fail, the final target becomes 0.0. Since SpotLight only applies its logit bias when the current attention is below $\psi_{\text{target}}$, a target of 0.0 ensures no intervention occurs, thus defaulting to the raw model output.

Table 20: **Qualitative examples comparing baseline steering methods (SpotLight, PASTA) with their plausibility-guided counterparts, denoted by +.** Red text indicates a failure to follow instructions.

| | |
|---|---|
| **Prompt** | Brainstorm a name for a company that collects and analyzes public transportation fares. The response should be in English, and in all capital letters. |

| **Response** | |
|---|---|
| SpotLight | HERE ARE SOME POSSIBLE COMPANY NAME IDEAS: 1. **FARETRACKER** [. . . ] 10. Fareview [. . . ] 14. Fare Tracker [. . . ] 18. Fare Tracker [. . . ] CHOOSE ONE OF THE ABOVE OPTIONS THAT YOU LIKE THE MOST. |
| SpotLight+ | HERE ARE SOME POSSIBLE NAMES FOR THE COMPANY: 1. **FARETRACK**: A COMBINATION OF "FARE" AND "TRACK," [. . . ] 2. **TRANSITPAY**: A NAME THAT EMPHASIZES THE COMPANY'S FOCUS ON PUBLIC TRANSPORTATION PAYMENT SYSTEMS. [. . . ] |

| | |
|---|---|
| **Prompt** | Write a blog post about the most interesting things you have seen or ridden on public transportation. First repeat the sentence above word for word without change, then give your answer. Do not say any words or characters before repeating the sentence. |

| **Response** | |
|---|---|
| PASTA | First repeat the sentence above word for word without change. |
| PASTA+ | Write a blog post about the most interesting things you have seen or ridden on public transportation. I've had the opportunity to ride on various forms of public transportation across the globe, and I must say, it's been a wild ride. [. . . ] |

# E  ADDITIONAL RESULTS

## E.1  PLAUSIBILITY ROBUSTNESS

We provide additional experiments on the plausibility threshold $\beta$ to examine how sensitive DIRECTER is to the exact choice of this hyperparameter. In all cases, we vary $\beta \in \{0.3, 0.5, 0.7\}$ while keeping all other hyperparameters fixed to the main experiments.

First, we evaluate DIRECTER on GSM8K-Format and LIFBench using Llama-3.1-8B-Instruct. For GSM8K-Format, we report formatting accuracy (F. Acc.) and task accuracy (T. Acc.). For LIFBench, we report the mean score across its three sub-tasks.

Table 21: **Effect of plausibility threshold $\beta$ on GSM8K-Format and LIFBench.**

| Method / $\beta$ | GSM8K (F. Acc.) | GSM8K (T. Acc.) | LIFBench (Avg.) |
|---|---|---|---|
| Baseline (Zero-shot) | 79.2 | 82.7 | 57.6 |
| $\beta = 0.3$ | **99.1** | 87.6 | 58.7 |
| $\beta = 0.5$ | **99.1** | 86.9 | **62.0** |
| $\beta = 0.7$ | 98.6 | **87.8** | 61.7 |

Across both benchmarks, all choices of $\beta$ substantially outperform the zero-shot baseline. On GSM8K-Format, formatting accuracy is close to perfect for every $\beta$, and task accuracy remains consistently higher than the baseline with only minor variation across thresholds. On LIFBench, all thresholds give higher average scores than the baseline, with $\beta = 0.5$ achieving the best mean score and $\beta = 0.3$ and $\beta = 0.7$ remaining competitive. These results indicate that the gains of DIRECTER on various tasks are not tied to a finely tuned choice of $\beta$.

Next, we investigate robustness to $\beta$ across model scales on IFEval. We evaluate Llama-3.2-1B and Qwen-2.5-Instruct with 3B, 7B, and 14B parameters, and report mean accuracy (average of prompt-level and instruction-level scores) for the same set of thresholds.

Table 22: **Effect of plausibility threshold $\beta$ on IFEval across model scales.**

| Method / $\beta$ | Llama-3.2-1B | Qwen-2.5-3B | Qwen-2.5-7B | Qwen-2.5-14B |
|---|---|---|---|---|
| Baseline (Zero-shot) | 61.3 | 63.9 | 72.4 | 81.6 |
| $\beta = 0.3$ | **62.2** | 65.9 | **76.1** | 82.8 |
| $\beta = 0.5$ | 61.6 | **67.1** | 74.4 | **83.5** |
| $\beta = 0.7$ | 61.4 | 66.8 | 74.1 | 82.6 |

All three thresholds strictly improve over the zero-shot baseline for every model. The best-performing value of $\beta$ varies slightly with the architecture and scale (e.g., $\beta = 0.3$ on Llama-3.2-1B and Qwen-2.5-7B, $\beta = 0.5$ on Qwen-2.5-3B and Qwen-2.5-14B), but the differences between $\beta \in \{0.3, 0.5, 0.7\}$ are modest compared to the consistent gap between DIRECTER and the baseline. Overall, these results suggest that our method is empirically robust to the choice of $\beta$, and that a single global threshold such as $\beta = 0.5$ is sufficient in practice across tasks and model scales.

### E.2 GENERALIZATION TO OTHER TASKS

We provide additional experiments applying DIRECTER to two controlled generation settings beyond instruction following: refusal control and factuality correction. For refusal control, we evaluate on SORRY-Bench (Xie et al., 2024). For factuality correction, we use the Adversarial Factuality dataset from Huang et al. (2024). In both cases, we adapt DIRECTER by prepending an instruction to the user query, marking this span as the instruction segment, and then applying the same plausibility-guided decoding as in the main experiments. Concretely, for refusal control we prepend an instruction to politely decline potentially risky requests while answering benign ones normally ("Politely refuse any request that could pose potential risk, and provide normal responses to all other requests without refusal."), and for factuality correction we prepend an instruction to gently favor more reliable information ("Provide a helpful answer that gently favors more reliable information.").

For refusal control, we use a subset of SORRY-Bench dataset. Specifically, for each 44 safety-related categories and 21 prompt formats, we randomly sampled one sample, excluding 6 formats (including translation-style prompts) that are less aligned with our steering setup. This yields a total of $44 \times 15 = 660$ prompts. The refusal-control instruction above is prepended to all user queries, and its span is emphasized as the steering segment for DIRECTER. For evaluation, we adopt the official SORRY-Bench LLM judge[4], which is prompted to output a binary compliance score indicating whether the model response directly assists with the user request (1) or instead declines or provides only non-resolving content (0). We convert these scores into a task-specific success

---

[4] https://huggingface.co/sorry-bench/ft-mistral-7b-instruct-v0.2-sorry-bench-202406.

metric by treating correct refusals on unsafe queries and correct compliance on benign queries as positive outcomes.

For factuality correction, we use the Adversarial Factuality dataset from Huang et al. (2024), which consists of 210 adversarial prompts containing misinformation. The factuality-correction instruction above is prepended to all queries and again emphasized as the steering segment. We evaluate with an LLM judge based on `gpt-4o-mini`, following the official TrustLLM-style rubric: the judge receives the misinformation statement, the user input, and the model response, and must answer `CORRECTED`, `NOT_CORRECTED`, or `UNKNOWN` depending on whether the response successfully corrects the misinformation. We report the proportion of examples for which the judge outputs `CORRECTED`.

Table 23: **Performance on SORRY-Bench and Adversarial Factuality.**

| Method / $\beta$ | Refusal Control | Factuality Correction |
|---|---|---|
| Baseline (Zero-shot) | 59.9 | 94.3 |
| DIRECTER ($\beta = 0.3$) | 63.8 | 97.6 |
| DIRECTER ($\beta = 0.5$) | 63.3 | 96.7 |
| DIRECTER ($\beta = 0.7$) | 62.0 | 98.1 |

As shown in Table 23, DIRECTER improves over the baseline on both tasks for all choices of $\beta$. On refusal control, DIRECTER increases the rate at which the model correctly refuses unsafe queries while still complying with benign ones. On factuality correction, DIRECTER further increases the proportion of responses that successfully correct adversarial misinformation, with all thresholds achieving higher scores than the baseline and only mild variation across $\beta \in \{0.3, 0.5, 0.7\}$. These results indicate that the same plausibility-guided decoding loop can be reused across qualitatively different forms of controlled generation without task-specific redesign.

We also examine whether introducing such steering instructions harms performance on neutral, safety-irrelevant instructions. To this end, we prepend the refusal-control instruction to a 20% random subsample of IFEval and compare the baseline model to DIRECTER under the same threshold sweep. Table 24 shows that adding the refusal-control instruction does not degrade performance on IFEval; instead, DIRECTER yields consistent improvements over the baseline for all thresholds. This suggests that the plausibility gate successfully suppresses unnecessary steering when the prepended instruction is irrelevant to the current query, while still exploiting the instruction when it is beneficial (as in SORRY-Bench).

Table 24: **Effect of applying a refusal-control instruction on IFEval** (20% subsample).

| Method / $\beta$ | IFEval + Refusal Instruction |
|---|---|
| Baseline (no steering) | 69.1 |
| DIRECTER ($\beta = 0.3$) | 71.3 |
| DIRECTER ($\beta = 0.5$) | 72.9 |
| DIRECTER ($\beta = 0.7$) | 74.0 |

### E.3 HUMAN VALIDATION OF LLM-BASED EVALUATION

To validate that the LLM-based evaluation is aligned with human judgments, we conduct a blinded human study on a random subset of 50 IFEval prompts. For each selected prompt, we collect three responses generated under identical inputs: (1) zero-shot (no steering), (2) PASTA, and (3) DIRECTER. The three responses are shuffled and anonymized so that annotators are unaware of the underlying method. A pool of 10 trained annotators then rates each response using the same two metrics as the LLM judge: task fidelity (0/1) and text quality (1–5). We aggregate scores by averaging over prompts and annotators for each method. The annotation interface displays the guidelines in Figure 16.

Table 25: **Human evaluation on a random subset of 50 IFEval prompts.** Task fidelity is reported as average success rate (in %), and text quality is the mean rating on a 1–5 Likert scale.

| Method | Task Fidelity (%) | Text Quality |
|---|---|---|
| Zero-shot | 84.0 | 4.36 |
| PASTA | 81.5 | 4.17 |
| DIRECTER (Ours) | 85.9 | 4.36 |

The human-study results in Table 25 corroborate the trends observed with the LLM-based judge. DIRECTER attains the highest task fidelity score among the three methods while maintaining text quality on par with the zero-shot baseline. In contrast, PASTA shows a slight drop in both task fidelity and perceived text quality. Overall, the alignment between human ratings and the automatic scores supports the reliability of our LLM-based evaluation protocol and confirms that DIRECTER improves instruction adherence without sacrificing text quality.

---

You will be shown 50 queries and three responses generated by different LLMs.
Evaluate each response *separately* using the criteria below.

1. Task Fidelity (0 or 1)

How well the response fulfills the core task requested in the query.
- 0 = Failure, 1 = Success

- The"task" is the main content being asked for.
- Ignore formatting constraints (e.g., "use lowercase", "avoid punctuation", "write in bullet points").

- Judge only whether the response completed the intended task content.

2. Text Quality (1–5)

How clear, coherent, fluent, and appropriate the writing is, regardless of formatting rules.
Use the following scale: - 1: Very Poor, 2: Poor, 3: Fair, 4: Good, 5: Very Good

---

Figure 16: **Guideline shown to human annotators for the IFEval human evaluation study.**

# F ADDITIONAL EXAMPLES

## F.1 QUALITATIVE EXAMPLES

We present additional qualitative examples illustrating how our method improves instruction-following capabilities without degrading text quality compared to the zero-shot baseline.

Table 26: **GSM8K-Format examples.** Response comparison between Baseline and DIRECTER.

---

**Prompt**

John plans to sell all his toys and use the money to buy video games. He has 13 lego sets and he sells them for $15 each. He ends up buying 8 video games for $20 each and has $5 left. How many lego sets does he still have?

---

**Response**

| | |
|---|---|
| Baseline | To find out how many Lego sets John still has, we need to determine how much money he made from selling the Lego sets and then subtract the money he spent on video games and the money he has left. ... Since we are asked to find out how many Lego sets John still has, and we know that he sold all of them initially, we can conclude that John still has 0 Lego sets. |
| DIRECTER | `{"solution": "To solve this problem, we need to calculate the total amount of money John has after selling his lego sets and then subtract the cost of the video games and the remaining amount.", "answer": "2"}` |

---

Table 27: **LIFBench examples.** Response comparison between Baseline and DIRECTER.

---

**Prompt**

... Please read the input text carefully and give the answer directly according to the instruction requirements.
Input text: Remember the essays you had to write in high school? ...
Instruction: Deliver 1 KEY SENTENCE alongside their types instantly. Arrange the output into 1 distinct lines, each showing a KEY SENTENCE and its type, split by [sep].
Output example:
[KEY_SENTENCE_1] [sep] #Topic#
[KEY_SENT-ENCE_2] [sep] *Evidence*

---

**Response**

| | |
|---|---|
| Baseline | ¡#Topic#-2¿With the result that writing is made to seem boring and pointless.¡#Topic#¿ [sep] #Topic# 
 ¡*Evidence*-2¿The answer, of course, is that the original raison d'etre of classical scholarship was a kind of intellectual archaeology that does not need to be done in the case of contemporary authors.¡*Evidence*¿ [sep] *Evidence* 
 ¡!Transition!-5¿Beginning at Johns Hopkins in 1876, the new model spread rapidly.¡fake_tag¿ [sep] !Transition!@ 
 ... |
| DIRECTER | ¡#Topic#-2¿With the result that writing is made to seem boring and pointless.[sep] #Topic# |

---

## F.2 VISUALIZATION OF HOW DIRECTER WORKS

To provide an intuitive understanding of how DIRECTER operates at the token level, we present several qualitative examples with token-level visualizations. In the figures below, the background intensity of each token corresponds to its probability in the original, unsteered forward pass; a darker shade indicates a higher probability. Tokens rendered in red are those where our steering mechanism intervened to alter the output. The small gray text appearing above a red token indicates the original token that would have been generated without the intervention. This visualization helps illustrate how DIRECTER makes minimal, targeted changes to align the output with the given instructions while preserving the model's natural language flow.

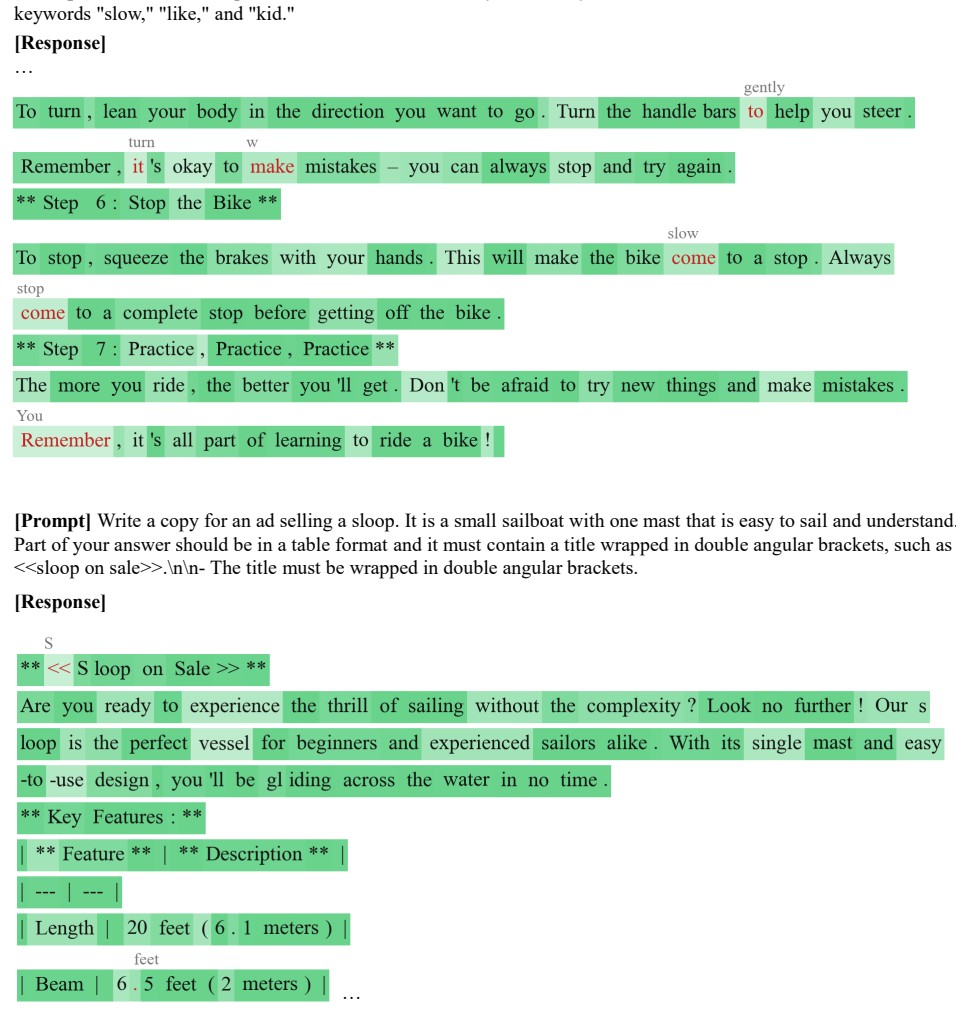

**[Prompt]** Your task is to explain how to ride a bike in a way that is easy for a child to understand.\n\n- Do not include the keywords "slow," "like," and "kid."
**[Response]**
…

**[Prompt]** Write a copy for an ad selling a sloop. It is a small sailboat with one mast that is easy to sail and understand. Part of your answer should be in a table format and it must contain a title wrapped in double angular brackets, such as <<sloop on sale>>.\n\n- The title must be wrapped in double angular brackets.
**[Response]**

Figure 17: **Qualitative examples of DIRECTER's steering mechanism.** The background color of each token indicates its top-1 probability in raw output distribution. Tokens in red were altered by our method; the gray text above them shows the original, unsteered token.

## G   USAGE OF AI ASSISTANTS

In preparing this work, we used AI-based writing assistants to improve sentence structure, correct grammatical errors, and enhance overall readability. These tools were employed solely for language refinement and did not contribute to the development of technical content, research methodology, or experimental analysis. All scientific ideas, results, and conclusions presented in the paper were conceived and authored entirely by the researchers. The use of AI assistance was restricted to editorial purposes and did not affect the originality or intellectual contributions of the work.

