# OpenReview forum: "Enhancing Instruction Following of LLMs via Activation Steering with Dynamic Rejection"
_ICLR.cc/2026/Conference — ICLR 2026 Poster_

### Official Review · Reviewer_Sn4M · 2025-10-26

**Soundness:** 4
**Presentation:** 4
**Contribution:** 3
**Rating:** 8
**Confidence:** 3

**Summary:**

This work introduces an attention-based steering mechanism for improved instruction following that maintains output quality and improves performance.

**Strengths:**

This work is well-written and scoped, and provides solid empirical evidence for their proposed control methodology. DIRECTER avoids the problem of "over-steering," which is present in other steering methods applied broadly to the input. The authors show that their method is successful across different benchmarks, model scales, and can be used successfully in conjunction with many different architectures. Importantly, they show that DIRECTER is efficient and performs well against simpler interventions such as plain prompting.

**Weaknesses:**

* Fixed plausibility threshold, while effective, is applied globally based on a hyperparameter sweep on a subset of the data, but it's unclear in the paper whether this generalizes across models or data domain types. A per-task adaptive value or assessment could help validate the applicability of this parameter.

* The LLM-based evaluation without human validation is a limitation, given the claims of text quality. Reliability of this as a metric should have a subset assessed manually to validate whether the text quality is maintained.

**Questions:**

1. How sensitive is the threshold to different domains or different models?

---

> ### Author Response · Authors · 2025-11-20
> **Response to Reviewer Sn4M (1/2)**
>
> Dear Reviewer Sn4M,
>
> We sincerely appreciate your thoughtful comments. We carefully considered and addressed the two issues you raised: (W1, Q1) sensitivity and generalization of the plausibility threshold $\beta$ across models and domains, including an option for per‑task adaptation; and (W2) validation of the LLM‑based evaluation with a human‑annotated subset to confirm that text quality is maintained. Below we provide detailed responses and precise manuscript edits.
>
> ---
>
> **[W1, Q1] Fixed plausibility threshold, while effective, is applied globally based on a hyperparameter sweep on a subset of the data, but it's unclear in the paper whether this generalizes across models or data domain types. A per-task adaptive value or assessment could help validate the applicability of this parameter.**
>
> Thank you for the thought comment. While we consider to use a fixed fixed plausibility threshold without any task-specific tuning to avoid additional cost for hyperparameter tuning, we agree that exploration for per-task adaptive value or assessment is valuable. To this end, we conducted additional experiments by varying $\beta$ across other datasets(GSM8K-Format, LIFBench) and different models(Llama-3.2-1B-Instruct, Qwen-2.5-Instruct-[3B, 7B, 14B]). The results below show that a single global $\beta=0.5$ perform well across domains and scales, while light tuning can yield marginal gains in a few settings (e.g., Qwen-2.5-7B-Instruct on IFEval).
>
> - **Experiments on other datasets**
>
> |  | GSM8K (F. Acc.) | GSM8K (T. Acc.) | LIFBench (Avg) |
> | --- | --- | --- | --- |
> | Baseline | 79.2 | 82.7 | 57.6 |
> | $\beta=0.3$ | 99.1 | 87.6 | 58.7 |
> | $\beta=0.5$ | 99.1 | 86.9 | 62.0 |
> | $\beta=0.7$ | 98.6 | 87.8 | 61.7 |
>
> - **Experiments across models on IFEval**
>
> |  | Llama 1B | Qwen 3B | Qwen 7B | Qwen 14B |
> | --- | --- | --- | --- | --- |
> | Baseline | 61.3 | 63.9 | 72.4 | 81.6 |
> | $\beta=0.3$ | 62.2 | 65.9 | 76.1 | 82.8 |
> | $\beta=0.5$ | 61.6 | 67.1 | 74.4 | 83.5 |
> | $\beta=0.7$ | 61.4 | 66.8 | 74.1 | 82.6 |
>
> These findings align with our robustness analysis in Figure 3a, which shows strong performance for a wide range of $\beta$ and cross-model results in Table 2. Together, this support the practicality of a global threshold while leaving room for optional per-task adjustment when desired. We have added these results and relevant discussions in Appendix E.3 of the revised draft.

---

> ### Author Response · Authors · 2025-11-20
> **Response to Reviewer Sn4M (2/2)**
>
> **[W2] The LLM-based evaluation without human validation is a limitation, given the claims of text quality. Reliability of this as a metric should have a subset assessed manually to validate whether the text quality is maintained.**
>
> Thank you for raising this important point. We use an LLM-based judge because it is a widely adopted, scalable and reproducible protocol for open-ended generation [1, 2, 3]. We specifically designed rubrics that separately scores text quality (1-5) and task fidelity (0 or 1). Also, we included the exact judging prompt to ensure reproducibility (Table 4 and Figure 8). This design intentionally decouples formatting constraints from response quality, making it an appropriate complement to our structured benchmarks.
>
> Nevertheless, we agree that an additional human validation is valuable to verify that these automated scores are indeed aligned with human judgments. To this end, we ran blinded human study on a random subset of IFEval prompts. 10 trained annotators compared zero-shot, PASTA, and DIRECTER responses under the same prompts and anonymization, using the same two metrics as the LLM judge: task fidelity (0/1) and text quality(1-5).
>
> The result below shows that human ratings corroborate the LLM-based evaluation; DIRECTER achieves the highest task fidelity while maintaining text quality on par with the non-steered baseline, whereas attention-level steering can degrade text quality under stricter adherence to instructions. We have added the full human-study table, the annotation guidelines, and details on sampling in Appendix E.5 of the revised draft.
>
> |  | Task Fidelity (%) | Text Quality |
> | --- | --- | --- |
> | Zero-shot | 84.0 | 4.36 |
> | PASTA | 81.5 | 4.17 |
> | DIRECTER | 85.9 | 4.36 |
>
> ---
>
> [1] Judging LLM-as-a-Judge with MT-Bench and Chatbot Arena (Zheng et al., 2023)
>
> [2] G-Eval: NLG Evaluation using GPT-4 with Better Human Alignment (Liu et al., 2023)
>
> [3] A Survey on LLM-as-a-Judge (Gu et al., 2024)
>
> ---
>
> We are grateful for these suggestions, which helped us strengthen the empirical comparison and clarify the rationale behind DIRECTER. We hope that our additional analyses and results satisfactorily address this concern.
>
> If you have any further questions, please let us know.
>
> Thank you very much,
>
> Authors

---

> > ### Comment · Reviewer_Sn4M · 2025-11-25
> >
> > Thank you for the response! I appreciate the addition of the human evaluations and urge the authors to include this in their revised manuscript; I will maintain my positive score.

---

> > > ### Author Response · Authors · 2025-11-26
> > > **Response to Reviewer Sn4M**
> > >
> > > We are happy to hear that our rebuttal addressed your concerns! As suggested, we will incorporate the human evaluation results (currently in Appendix E.5) into the main manuscript in the revised version. Please don't hesitate if you have any further questions.

---

### Official Review · Reviewer_CXSA · 2025-10-26

**Soundness:** 3
**Presentation:** 2
**Contribution:** 3
**Rating:** 4
**Confidence:** 3

**Summary:**

The paper proposes DIRECTER, a dynamic activation-steering method that improves instruction-following of LLMs by adaptively adjusting during decoding. The main idea is a plausibility-guided decoding loop that compares steered and raw token distributions at each step. If the steered token is deemed implausible, steering strength is reduced by halving the set of “steered layers,” ranked by a precomputed attention sensitivity score.

**Strengths:**

1. The plausibility-guided decoding loop is conceptually simple yet effective. Its dynamic halving mechanism mimics reinforcement learning–style adaptive control but without extra training.
2. The method introduces minimal memory overhead and modest throughput reduction (≈ 16%)
3. The extensive ablation and robustness analyses (Fig. 2–3) convincingly demonstrate the benefits of dynamic steering and the effectiveness of layer ranking.

**Weaknesses:**

1. The attention sensitivity metric (Eq. 3–4) is ad-hoc and lacks theoretical justification. The “direct” and “propagated” effects are computed via cosine distance differences, but no intuition or derivation is provided. Why does summing cosine-distance deviations across layers accurately capture “influence”?
2. The method section is hard to follow due to poor narrative flow, cross references and undefined notation. For example, Readers are told what each symbol means after it’s used — e.g., $𝐿_{cand,𝑡}$ appears in the plausibility rule before explaining how it’s initialized or updated. The paper didn't provides a clear figure or algorithm before diving into formulas. It is better to summarize their method in a pseudocode algorithm.
3. Eq. (2) assumes top-1 plausibility comparison, but this is sensitive to token-level noise. It could be better if they evaluate with multiple thresholds $\beta \in [0.3, 0.5, 0.7]$ to show sensitivity (only $\beta = 0.5$ is used).

**Questions:**

* Have you tried applying DIRECTER to other forms of controlled generation (e.g., factuality correction, refusal control)? A small-scale results would demonstrate the method’s broader applicability and strengthen the paper’s impact claim that DIRECTER is a “general mechanism” for mitigating oversteering.

---

> ### Author Response · Authors · 2025-11-20
> **Response to Reviewer CXSA (1/5)**
>
> Dear Reviewer CXSA,
>
> We sincerely appreciate your thoughtful comments. We carefully considered and addressed the issues you raised: (W1) the intuition and justification behind the attention‑sensitivity metric in Eq. 3-4, (W2) clarity and organization of the method section, (W3) robustness of the top‑1 plausibility threshold with a sensitivity analysis over $\beta$, and (Q1) applicability of DIRECTER to other controlled‑generation settings. Below we provide detailed responses and precise manuscript edits.

---

> ### Author Response · Authors · 2025-11-20
> **Response to Reviewer CXSA (2/5)**
>
> **[W1] The attention sensitivity metric (Eq. 3–4) is ad-hoc and lacks theoretical justification. The “direct” and “propagated” effects are computed via cosine distance differences, but no intuition or derivation is provided. Why does summing cosine-distance deviations across layers accurately capture “influence”?**
>
> Thank you for the opportunity to clarify the rationale behind our metric. We summarize the motivation in three points:
>
> **(1) Why separate “direct” and “propagated” effects, and why sum them across layers?**
>
> First, we emphasize that estimating the downstream impact of steering multiple layers is inherently difficult because perturbations accumulate non-linearly. Prior work such as PASTA addresses this by using auxiliary data to evaluate layer importance, but this introduces additional pre-compute cost and potential overfitting. Our goal is to estimate layer sensitivity *without* relying on external data.
>
> To make this complex problem be simpler, we approximate the overall steering effect by isolating each layer’s marginal contribution and then summing across layers. However, simply measuring the local change at the steered layer is not enough as steering also alters the *inputs* to all subsequent layers. Therefore, we designed our metric to capture these two distinct effects as detailed in Eq. 3: (1) the direct impact on the steered layer's attention output, and (2) the propagated impact on the attention input of other layers. Summing these shifts over layers provides a practical proxy for the global influence of steering at each layer.
>
> This approximation is empirically validated: as shown in Table 3, reversing or randomizing the ranking substantially degrades performance, compared to using the obtained layer ranking in this way. Appendix D.2 further compares variants of the metric (e.g., using only direct or only propagated terms) and supports the full formulation.
>
>
>
> **(2) Why cosine distance?**
>
> We use cosine distance because our steering method scales key vectors (Eq. 1), thereby modifying *attention weights* after the softmax. Namely, this steering changes the coefficients for value vectors and hence it is more relevant to directional shift. Therefore, we choose cosine distance as cosine distance explicitly measures directional deviation, making it a suitable proxy for semantic change. In contrast, norm-based metrics like L2 distance is dominated by magnitude changes that need not correlate with meaningful directional shifts in the representation.
>
> This choice also aligns with KV cache compression work [1, 2], where use cosine similarity to identify redundant layers where input-output representations remain stable, allowing them to be discarded with minimal impact. We apply the inverse logic: we seek layers where a steering intervention produces the maximum directional shift, signaling high steering leverage.
>
> To further support this choice of cosine metric, we conducted an additional experiment comparing our cosine-based ranking against an L2-norm-based ranking. The result below shows that ranking by cosine distance consistently yields higher instruction-following accuracy than L2-based one, validating that capturing directional sensitivity with cosine distance is the more accurate predictor of steering effect.
>
> |  | IFEval (Avg Acc.) | GSM8K-Format (F. Acc. / T. Acc.) | LIFBench (Avg Acc.) |
> | --- | --- | --- | --- |
> | Baseline | 77.5 | 79.2 / 82.7 | 57.6 |
> | L2 based | 80.5 | 96.1 / 83.2 | 58.6 |
> | Cosine based (DIRECTER) | 81.8 | 99.1 / 86.9 | 62.0 |
>
>
>
> **(3) A more central contribution: plausibility-guided decoding loop**
>
> Finally, we emphasize that while sensitivity-based ranking offers the best results, our plausibility-guided decoding loop remains effective even with simpler layer-selection strategies. For instance, randomly halving layers at each iteration still improves performance (e.g., Acc. on IFEVal subset: 77.5 → 80.2 in Table 3). Namely, the ranking procedure is a modular component that can be replaced or further improved in future work.
>
> Thank you again for the insightful feedback. We have added this extended explanation and the L2-ablation results to Appendix E.6, and will incorporate the key intuition into the main text in the camera-ready version.
>
> ---
>
> [1] SqueezeAttention: 2D Management of KV-Cache in LLM Inference via Layer-wise Optimal Budget (Wang et al., 2024)
>
> [2] Scissorhands: Exploiting the Persistence of Importance Hypothesis for LLM KV Cache Compression at Test Time (Liu et al., 2023)

---

> ### Author Response · Authors · 2025-11-20
> **Response to Reviewer CXSA (3/5)**
>
> **[W2] The method section is hard to follow due to poor narrative flow, cross references and undefined notation. For example, Readers are told what each symbol means after it’s used — e.g., $L_\text{cand,t}$ appears in the plausibility rule before explaining how it’s initialized or updated. The paper didn't provides a clear figure or algorithm before diving into formulas. It is better to summarize their method in a pseudocode algorithm.**
>
> Thank you for constructive comments to improve the readability of our manuscript. We first clarify that the original manuscript already includes a figure of high-level overview (Figure 1) and a full pseudocode description (Algorithm 1 in Appendix C.1). In addition, $\mathcal L_\text{ranked}$ and $\mathcal L_{\text{cand},t}$ are defined and initialized (lines 183-189) before the plausibility rule is introduced (line 190). However, we agree that the method section could be presented more clearly with the following edits and we incorporate these in the revised manuscript:
>
> - Correction of inconsistent notation in Line 171, replacing the earlier use of $S$ with $\mathcal L$.
> - Referring Figure 1 and Algorithm 1 at the beginning of Section 3.2 as follow: (Lines 177-178) “*For a high-level overview and the full procedure, see Figure 1 and Algorithm 1, respectively.*”
>
> We will further make effort to improve the narrative flow and make the definitions and references easy to follow in method section. We appreciate your comment again, it has been very helpful for improving clarity of the paper.

---

> ### Author Response · Authors · 2025-11-20
> **Response to Reviewer CXSA (4/5)**
>
> **[W3] Eq. (2) assumes top-1 plausibility comparison, but this is sensitive to token-level noise. It could be better if they evaluate with multiple thresholds $\beta\in[0.3,0.5,0.7]$ to show sensitivity (only $\beta=0.5$  is used).**
>
> Thank you for the insightful comment regarding the evaluation with plausibility threshold $\beta$. We first want to clarify that sensitivity analysis for $\beta$ (on IFEval under Llama-3.1-8B-Instruct) was already presented in Figure 3a of the original draft. This result shows that our method generalizes across wide range of $\beta$.
>
> Nevertheless, to further validate the robustness of $\beta$ across various setups, we conduct additional experiments on other datasets (GSM8K-Format and LIFBench in Table 1) and models (Llama-3.2-1B-Instruct, Qwen-2.5-Instruct-[3B,7B,14B] in Table 2) by varying the $\beta$ among $[0.3, 0.5, 0.7]$.
>
> - **Experiments on other datasets**
>
> |  | GSM8K (F. Acc.) | GSM8K (T. Acc.) | LIFBench (Avg) |
> | --- | --- | --- | --- |
> | Baseline | 79.2 | 82.7 | 57.6 |
> | $\beta=0.3$ | 99.1 | 87.6 | 58.7 |
> | $\beta=0.5$ | 99.1 | 86.9 | 62.0 |
> | $\beta=0.7$ | 98.6 | 87.8 | 61.7 |
>
> - **Experiments across models on IFEval**
>
> |  | Llama 1B | Qwen 3B | Qwen 7B | Qwen 14B |
> | --- | --- | --- | --- | --- |
> | Baseline | 61.3 | 63.9 | 72.4 | 81.6 |
> | $\beta=0.3$ | 62.2 | 65.9 | 76.1 | 82.8 |
> | $\beta=0.5$ | 61.6 | 67.1 | 74.4 | 83.5 |
> | $\beta=0.7$ | 61.4 | 66.8 | 74.1 | 82.6 |
>
> The results above show that DIRECTER consistently improves the performance, supporting our claim that DIRECTER is not sensitive to the exact choice of $\beta$ and generalize across tasks and model architectures. We have added these results in Appendix E.3 of the revised draft.

---

> ### Author Response · Authors · 2025-11-20
> **Response to Reviewer CXSA (5/5)**
>
> **[Q1] Have you tried applying DIRECTER to other forms of controlled generation (e.g., factuality correction, refusal control)? A small-scale results would demonstrate the method’s broader applicability and strengthen the paper’s impact claim that DIRECTER is a “general mechanism” for mitigating oversteering.**
>
> Thank you for the thoughtful comments! Indeed, DIRECTER is easily applicable for various controlled generation scenarios, and we totally agree that additionally showing this aspect is helpful to demonstrate the effectiveness of DIRECTER. Following your suggestion, we conducted additional experiments to test our method on two new tasks, refusal control (subsampled SORRY-Bench [1]) and factuality correction (Adversarial Factuality [2]). To apply DIRECTER for each task, we prepended instructions to refuse query which potentially cause risk (`Politely refuse any request that could pose potential risk, and provide normal responses to all other requests without refusal.`) or to respond with reliable information (`Provide a helpful answer that gently favors more reliable information.`), respectively. We report Refusal rate(%) for refusal control task and Correction percentage (%) for factuality correction task. Both are judged by LLM, following the official implementation. The results are presented in the below table. Here, we confirmed that DIRECTER still generalizes well to these new tasks.
>
> |  | Refusal Control | Factuality Correction |
> | --- | --- | --- |
> | Baseline | 59.9 | 94.3 |
> | DIRECTER ($\beta=0.3$) | 63.8 | 97.6 |
> | DIRECTER ($\beta=0.5$) | 63.3 | 96.7 |
> | DIRECTER ($\beta=0.7$) | 62.0 | 98.1 |
>
> Next, to further validate that this approach (DIRECTER with added instruction) do not compromise performance on neutral query, we conducted additional experiment by prepending the same refusal instruction on subsampled IFEval dataset (20% random selection). The results below shows that adding such steering instruction does not harm performance when that  instruction is not relevant to the query.
>
> |  | Mean Acc. (%) on IFEval + Refusal Instruction |
> | --- | --- |
> | Baseline | 69.1 |
> | DIRECTER ($\beta=0.3$) | 71.3 |
> | DIRECTER ($\beta=0.5$) | 72.9 |
> | DIRECTER ($\beta=0.7$) | 74.0 |
>
> Finally, we would like to highlight the results in Figure 2b which demonstrates the compatibility of DIRECTER’s idea to mitigate oversteering in other steering baselines. Here, we showed that prior steering methods often oversteer when hyperparameters are not carefully tuned to the target task. However, by applying the plausibility-guided decoding loop of DIRECTER, the issue is alleviated and performance improves over the zero-shot baseline. For example, 72.8% → 78.2% for PASTA, and 68.8% → 79.3% for SpotLight on IFEval benchmark.
>
> Overall, with these results, we believe that DIRECTER can serve as a general mechanism for mitigating oversteering in various setups. We also added these results with detailed setup and discussion in Appendix E.4.
>
> ---
>
> [1] SORRY-Bench: Systematically Evaluating Large Language Model Safety Refusal (Xie et al., 2024)
>
> [2] TrustLLM: Trustworthiness in Large Language Models (Huang et al., 2024)
>
> ---
>
> We are grateful for these suggestions, which helped us strengthen the empirical comparison and clarify the rationale behind our attention-sensitivty based layer ranking. We hope that our additional analyses and results satisfactorily address this concern.
>
> If you have any further questions, please let us know.
>
> Thank you very much,
>
> Authors

---

> > ### Comment · Reviewer_CXSA · 2025-11-22
> >
> > Thanks for the explanation and extensive experiment results.
> > My concerns have been addressed. I updated my score accordingly.

---

> > > ### Author Response · Authors · 2025-11-22
> > > **Response to Reviewer CXSA**
> > >
> > > We are glad to see that our rebuttal successfully clarified your questions! Please let us know if you have any further questions.

---

### Official Review · Reviewer_tbf4 · 2025-10-30

**Soundness:** 3
**Presentation:** 3
**Contribution:** 2
**Rating:** 4
**Confidence:** 3

**Summary:**

This paper introduces DIRECTER , a novel inference-time activation steering method that enhances the instruction-following ability of large language models . Unlike existing static steering methods which risk oversteering，DIRECTER adaptively adjusts the steering strength at each decoding step through a plausibility-guided decoding loop. It scales the key-value  cache layers dynamically based on attention sensitivity analysis, ensuring plausible outputs while mitigating instruction misalignment. Experiments on IFEval, LIFBench, and GSM8K-Format show consistent improvements (up to +6.5%) in instruction accuracy without sacrificing fluency or reasoning performance. The method is model-agnostic and incurs modest computational overhead (~16% latency).

**Strengths:**

- The paper accurately identifies the "oversteering" problem in existing activation steering techniques. The proposed DIRECTER provides a dynamic, adaptive control mechanism rather than relying on static, manually-tuned hyperparameters.
- The results strongly demonstrate the method's key advantage: it significantly improves instruction following without sacrificing core task accuracy or text quality.
- The evaluation is thorough, using diverse benchmarks including IFEval (strict instructions), LIFBench (long context), and the newly-created GSM8K-Format.

**Weaknesses:**

- RQ2 requires  the generalization of DIRECTER across different architectures and model sizes. To make the comparison more complete, Table 2 should also include results of other steering-based methods such as PASTA and SpotLight on more LLM architecture (e.g., Qwen) .
- The choice of the probability-ratio–based plausibility criterion (Eq. 2) is currently mainly empirical. However, commonly used measures for comparing probability distributions include KL divergence or JS divergence. The authors should compare these alternatives to justify Eq. (2).
- The method proposed by the authors shows some improvements in text quality and task performance. However, from an efficiency perspective, the overhead is prohibitive. In particular, the time required to generate the first token is roughly five times of SpotLight.

**Questions:**

Please refer to weaknesses.

---

> ### Author Response · Authors · 2025-11-20
> **Response to Reviewer tbf4 (1/3)**
>
> Dear Reviewer tbf4,
>
> We sincerely appreciate your thoughtful comments. We carefully considered and addressed the three issues you raised: (W1) broader generalization and comparisons, (W2) justification of the plausibility criterion in Eq. (2) versus KL/JS divergence alternatives, and (W3) efficiency concern on out method. Below we provide detailed response and precise manuscript edits.
>
> ---
>
> **[W1] RQ2 requires the generalization of DIRECTER across different architectures and model sizes. To make the comparison more complete, Table 2 should also include results of other steering-based methods such as PASTA and SpotLight on more LLM architecture (e.g., Qwen) .**
>
> Thank you for the constructive comment. We agree that including additional results for other steering-based methods across different architectures strengthens the evidence for RQ2.
>
> Following your suggestion, we evaluated PASTA and SpotLight on the same set of models used in Table 2, using the same hyperparameters selected in Table 1 to ensure a fair comparison with DIRECTER. The additional results (shown below) further confirm the generalizability of DIRECTER: in most cases, DIRECTER outperforms other steering baselines with large margins, whereas the baselines frequently underperform even the non-steered (zero-shot) setting due to oversteering. We present these additional results and the corresponding discussion in Appendix E.2 of the revised draft, and will incorporate them into main draft of the camera ready version.
>
> |  | Llama3.2 1B | Qwen2.5 3B | Qwen2.5 7B | Qwen2.5 14B | Avg Rank. |
> | --- | --- | --- | --- | --- | --- |
> | Baseline | 61.3 | 63.9 | 72.4 | 81.6 | 3.00 (3) |
> | PASTA | 59.7 | 65.2 | 73.0 | 80.1 | 3.25 (4) |
> | SpotLight | 60.6 | 62.8 | 74.9 | 81.7 | 2.50 (2) |
> | DIRECTER | 61.6  | 67.1 | 74.4 | 83.5 | 1.25 (1) |

---

> ### Author Response · Authors · 2025-11-20
> **Response to Reviewer tbf4 (2/3)**
>
> **[W2] The choice of the probability-ratio–based plausibility criterion (Eq. 2) is currently mainly empirical. However, commonly used measures for comparing probability distributions include KL divergence or JS divergence. The authors should compare these alternatives to justify Eq. (2).**
>
> Thank you for the insightful suggestion. While KL and JS divergence are standard for comparing probability distributions as the reviewer pointed out, they present several issues for our plausibility-guided decoding loop.
>
> First, KL (or JS) divergence measures aggregate shifts over the whole vocabulary, which means small diffuse changes in the long tail region (with low-probability tokens) can inflate this divergence even when the top-1 token remains acceptable, leading to false rejections.
>
> Moreover, as KL divergence has no fixed global scale, varying across models and even contexts, setting a single global threshold is brittle. Additionally, KL/JS divergence is not compatible with our top-2 probability based skipping mechanism (denoted in lines 195-201), which is designed to lower unnecessary computations.
>
> Nonetheless, we agree that the suggested proposed comparison would further validate our method design. To this end, we first fit proper thresholds for KL/JS divergence on 20% subsampled IFEval dataset. Then, we ran experiments with another subset three times. The results below shows KL and JS divergence-based criterion underperformed our method (Eq.2) and were more sensitive to threshold choice. We added these new results and a brief discussion in Appendix E.1 of the revised draft.
>
> | Method | Value | Accuracy (%) |
> | --- | --- | --- |
> | Baseline | - | $72.6 \pm1.99$ |
> | Probability ratio ($\beta$) | 0.3 | $75.7 \pm 0.53$ |
> |  | 0.5 | **$78.2 \pm 0.82$** |
> |  | 0.7 | $77.6 \pm 2.11$ |
> | KL divergence ($\tau_\text{KL}$) | 0.1 | $72.8 \pm 0.46$ |
> |  | 0.2 | $74.3 \pm 1.20$ |
> |  | 0.3 | $74.3 \pm 2.01$ |
> | JS divergernce ($\tau_\text{JS}$) | 0.05 | $74.8 \pm 0.85$ |
> |  | 0.10 | $72.6 \pm 2.15$ |
> |  | 0.15 | $74.6 \pm 2.94$ |
>
> On the other hand, we note that this probability-ratio-based criterion also aligns well with previous works, such as contrastive decoding framework, which control generation by comparing top-token preferences across distributions or models. For example, [1, 2, 3] also leverage the probability of selected token (top-1 token) is reference distribution and modified distribution to apply plausibility constraint.
>
> ---
>
> [1] Contrastive Decoding: Open-ended Text Generation as Optimization (Lisa et al., 2022)
>
> [2] DoLa: Decoding by Contrasting Layers Improves Factuality in Large Language Models (Chaung et al., 2023)
>
> [3] SafeDecoding: Defending against Jailbreak Attacks via Safety-Aware Decoding (Xu et al., 2024)

---

> ### Author Response · Authors · 2025-11-20
> **Response to Reviewer tbf4 (3/3)**
>
> **[W3] The method proposed by the authors shows some improvements in text quality and task performance. However, from an efficiency perspective, the overhead is prohibitive. In particular, the time required to generate the first token is roughly five times of SpotLight.**
>
> Thank you for providing the opportunity to clarify this efficiency concern. We clarify that the overall computational overhead of DIRECTER is NOT prohibitive. As shown in Figure 4, in end-to-end generation, it is more efficient than prior steering baselines (2x faster than SpotLight) and even remains close to the zero-shot setting, introducing modest $\approx16\%$ latency.
>
> The higher time to first token (TTFT), pointed by the reviewer, indeed stems from a one-time layer-sensitivity ranking performed after prefill. At cost of TTFT, it does not require extra data for layer ranking analysis like PASTA, which requires $N\times L\times H$ forward passes for head profiling (where $N, L, H$ each indicates number of data, layers, heads respectively). Also, we remark that one can easily reduce this cost by substituting this with other layer sensitivity analysis methods. More importantly, after this process, DIRECTER only requires minimal, necessary computation for steering, leveraging the efficient skipping mechanism. However other methods like SpotLight effectively doubles softmax operations in all decoding steps which incur higher latency. This is further elaborated in Appendix D.4 and Figure 12, which shows DIRECTER is still efficient in possibly the worst case, where generated token length is short (<100).
>
> Finally, as our method applies steering to KV cache, it remains compatible with FlashAttention, whereas attention-level methods like SpotLight and PASTA are not, requiring explicit materialization of attention matrices. This compatibility also provides the additional advantages in the computational perspective.
>
> ---
>
> We are grateful for these suggestions, which helped us strengthen the empirical comparison and clarify the rationale behind Eq. (2). We hope that our additional analyses and results satisfactorily address this concern.
>
> If you have any further questions, please let us know.
>
> Thank you very much,
>
> Authors

---

> ### Author Response · Authors · 2025-11-26
> **A Gentle Invitation for Further Discussion**
>
> Dear Reviewer tbf4,
>
> Again, we would like to kindly express our appreciation for your thoughtful reviews and the time you have devoted to evaluating our submission.
>
> If you have any additional thoughts or follow-up comments after reading our author response, we would be very grateful to hear them. Your insights during the discussion phase are extremely valuable and help us further clarify and strengthen the work.
>
> Thank you very much for your time and engagement.
>
> Kindest regards,
> Authors

---

### Author Response · Authors · 2025-11-20
**General Response**

Dear reviewers and AC,

We sincerely appreciate your valuable time and effort spent reviewing our manuscript. As reviewers highlighted, we propose a **conceptually simple yet effective (CXSA)** and **novel (tbf4)** inference-time activation steering method for improving instruction following, that shows **strong empirical evidence (Sn4M)** and **consistent improvements (tbf4)** across diverse benchmarks with **extensive ablation and robustness analyses (CXSA)**. In response to the comments, we have carefully revised and enhanced the manuscript, including the following additional discussions and experiments:

- Enhanced method description and improved notation (Section 3.2)
- Comparative analysis of probability-ratio criterion versus KL/JS divergence (Appendix E.1)
- Generalization to diverse architectures and model sizes (Appendix E.2)
- Robustness analysis of the plausibility threshold across benchmarks and models (Appendix E.3)
- Application of DIRECTER to Refusal Control and Factuality Correction tasks (Appendix E.4)
- Human validation of the LLM-based evaluation to ensure text quality (Appendix E.5)
- Justification and empirical validation of the attention sensitivity metric (Appendix E.6)

In the revised manuscript, these updates are temporarily highlighted in $\textbf{\color{magenta}pink}$ for your convenience to check.

We sincerely believe that these updates may help us better deliver the benefits of the proposed DIRECTER to the ICLR community.

Thank you very much,

Authors.

---

### Meta-Review · Area_Chair_W5KL · 2026-01-06

**Summary:**

This work introduces a technique to improve LLM instruction-following via steering, but in a way that guards against *over*steering via rejection sampling (which is informed by similarity to the counterfactual, unsteered output). Reviewers generally found this idea intuitive, and the empirical analysis comprehensive and compelling; the expanded set of results (responsive to the initial round of reviews) should strengthen this impression.

**Reviewer Concerns:**

Concerns about including additional points of comparison (tbf4) were addressed in rebuttal. Other points of concern were mostly about motivation and intuition beyond methods, which the authors have also made an attempt to provide.

**Reviewer Scores:**

CXSA explicitly did update their score in light of additional results provided.

tbf4 did not engage in discussion but it appears the authors have adequately addressed their concerns.

Sn4M was already positive and acknowledged the author response satisfied their outstanding question.

---

### Decision · Program_Chairs · 2026-01-26

Accept (Poster)